# Asymmetric Perturbation in Solving Bilinear Saddle-Point Optimization

## Abstract

This paper proposes an asymmetric perturbation technique for solving bilinear saddle-point optimization problems, commonly arising in minimax problems, game theory, and constrained optimization. Perturbing payoffs or values is known to be effective in stabilizing learning dynamics and equilibrium computation. However, it requires careful adjustment of the perturbation magnitude; otherwise, learning dynamics converge to only an approximate equilibrium. To overcome this, we introduce an asymmetric perturbation approach, where only one player's payoff function is perturbed. Exploiting the near-linear structure of bilinear problems, we show that, for a sufficiently small perturbation, the equilibrium strategy of the asymmetrically perturbed game coincides with an equilibrium strategy of the original game. This property yields a perturbation-based learning algorithm that achieves convergence to an equilibrium strategy in the original game without requiring parameter adjustments. Furthermore, we empirically demonstrate fast convergence toward equilibria in both normal-form and extensive-form games.

## 1 Introduction

This paper proposes an asymmetric perturbation technique for solving saddle-point optimization problems, commonly arising in minimax problems, game theory, and constrained optimization. Over the past decade, no-regret learning algorithms have been extensively studied for computing (approximate) solutions or equilibria. When each player minimizes regret, the time-averaged strategies approximate Nash equilibria in two-player zero-sum games; that is, *average-iterate convergence* is guaranteed. However, the actual sequence of strategies does not necessarily converge and can cycle or diverge even in simple bilinear cases (Mertikopoulos et al., 2018; Bailey & Piliouras, 2018; Cheung & Piliouras, 2019). This is problematic, especially in large-scale games with neural network policies, since averaging requires storing a separate model at every iteration.

This motivates the study of *last-iterate convergence*, a stronger notion than average-iterate convergence, in which the strategies themselves converge to an equilibrium. One successful approach is to use *optimistic* learning algorithms, which essentially incorporate a one-step optimistic prediction that the environment will behave similarly in the next step. This idea has led to several effective algorithms, including Extra-Gradient methods (EG) (Liang & Stokes, 2019; Mokhtari et al., 2020), Optimistic Gradient Descent Ascent (OGDA) (Daskalakis & Panageas, 2019; Gidel et al., 2019; Mertikopoulos & Zhou, 2019), and Optimistic Multiplicative Weights Update (OMWU) (Daskalakis & Panageas, 2019; Lei et al., 2021a). However, in large-scale settings where the gradient must be estimated from data or simulation, these algorithms can lose the last-iterate convergence property. For example, Abe et al. (2022) reports empirical non-convergence behavior under bandit feedback.

Alternatively, perturbing the payoffs with strongly convex penalties (Facchinei & Pang, 2003) has long been recognized as an effective technique for achieving last iterate convergence (Koshal et al., 2010; Tatarenko & Kamgarpour, 2019). This line of work has also shown strong performance in practical settings, including learning in large-scale games (Bakhtin et al., 2023) and fine-tuning large language models via preference optimization (Ye et al., 2024), often in place of optimistic algorithms. In prior work, the perturbation is almost always applied *symmetrically*, meaning both players' payoffs are augmented with the same strongly convex penalty, meaning both players' payoff functions are perturbed by a strongly convex penalty. A known limitation is that, with a fixed perturbation strength, the solution remains only an approximation of the original game's equilibrium, and the deviation

scales with the strength of the perturbation (Liu et al., 2023; Abe et al., 2024). Consequently, practice typically uses either a decreasing schedule or a horizon-dependent small value for the perturbation strength, both of which require careful hyperparameter tuning.

To avoid these restrictions, we develop an *asymmetric perturbation* approach that requires no careful hyperparameter tuning or scheduling for the perturbation strength. In this scheme, only one player's payoff function is perturbed while the other remains unperturbed. This simple modification yields a qualitatively different outcome. For any sufficiently small perturbation strength within a broad and practical range, the equilibrium strategy of the perturbed game coincides with that of the original game (see Theorem 3.1). Intuitively, leaving player $y$ unperturbed preserves the linearity of player $x$'s original objective, so adding a strongly convex perturbation does not significantly shift the solution (see Figure 2). Consequently, solving the asymmetrically perturbed game suffices to recover an equilibrium strategy of the original game.

Furthermore, to demonstrate the effectiveness of our findings, we provide two applications in normal-form and extensive-form games. First, we incorporate the technique into a gradient-based learning algorithm, which provably converges to a saddle point with a guaranteed rate (see Theorem 4.1 and Corollary 4.2). Empirical results on benchmark normal-form games show accelerated convergence. Second, we apply the technique to Counterfactual Regret Minimization (CFR) (Zinkevich et al., 2007), a widely used method in extensive-form games, and demonstrate significant improvements in convergence speed on standard benchmarks. While our analysis focuses on bilinear games, the structural insight behind the asymmetric perturbation may extend beyond this setting, including two-player zero-sum Markov games, and serves as a bridge to the design of new perturbation-based learning algorithms.

## 2 PRELIMINARIES

**Bilinear saddle-point optimization problems.** In this study, we focus on the following bilinear saddle-point problem:

$$\min_{x \in \mathcal{X}} \max_{y \in \mathcal{Y}} x^\top A y, \tag{1}$$

where $\mathcal{X} \subseteq \mathbb{R}^m$ (resp. $\mathcal{Y} \subseteq \mathbb{R}^n$) represents the $m$-dimensional (resp. $n$-dimensional) convex strategy space for player $x$ (resp. player $y$), and $A \in \mathbb{R}^{m \times n}$ is a game matrix. We assume that $\mathcal{X}$ and $\mathcal{Y}$ are polytopes. We refer to the function $x^\top A y$ as the *payoff function*, and write $z = (x, y)$ as the *strategy profile*. This formulation includes many well-studied classes of games, such as two-player normal-form games and extensive-form games with perfect recall[1].

**Nash equilibrium.** This study aims to compute a minimax or maximin strategy in the optimization problem Eq. (1). Let $\mathcal{X}^* := \arg\min_{x \in \mathcal{X}} \max_{y \in \mathcal{Y}} x^\top A y$ denote the set of minimax strategies, and let $\mathcal{Y}^* := \arg\max_{y \in \mathcal{Y}} \min_{x \in \mathcal{X}} x^\top A y$ denote the set of maximin strategies. It is well-known that any strategy profile $(x^*, y^*) \in \mathcal{X}^* \times \mathcal{Y}^*$ is a *Nash equilibrium*, which satisfies the following condition:

$$\forall (x, y) \in \mathcal{X} \times \mathcal{Y}, \ (x^*)^\top A y \leq (x^*)^\top A y^* \leq x^\top A y^*.$$

Based on the minimax theorem (v. Neumann, 1928), every equilibrium $(x^*, y^*) \in \mathcal{X}^* \times \mathcal{Y}^*$ attains the identical value, denoted as $v^*$, which can be expressed as:

$$v^* := \min_{x \in \mathcal{X}} \max_{y \in \mathcal{Y}} x^\top A y = \max_{y \in \mathcal{Y}} \min_{x \in \mathcal{X}} x^\top A y.$$

We refer to $v^*$ as the *game value*. To quantify the proximity to equilibrium for a given strategy profile $(x, y)$, we use *NashConv*, which is defined as follows:

$$\text{NashConv}(x, y) = \max_{\tilde{y} \in \mathcal{Y}} x^\top A \tilde{y} - \min_{\tilde{x} \in \mathcal{X}} (\tilde{x})^\top A y.$$

---

[1]Two-player extensive-form games with perfect recall can be expressed as bilinear problems using sequence-form strategies (von Stengel, 1996)

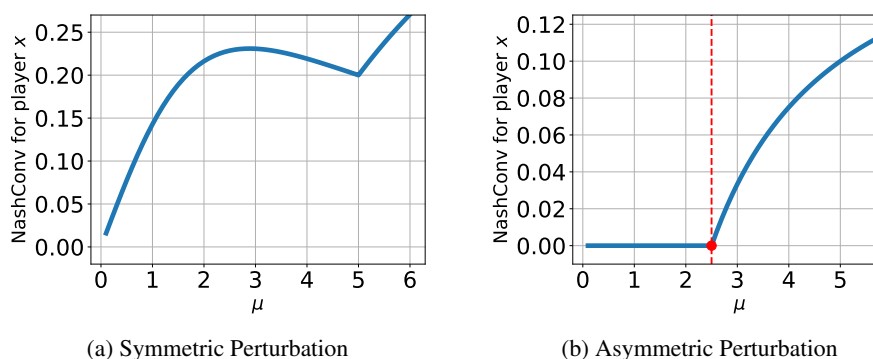

(a) Symmetric Perturbation  (b) Asymmetric Perturbation

Figure 1: The proximity of $x^\mu$ to $x^*$ under the symmetric perturbation and the asymmetric perturbation with varying $\mu$. The game matrix $A$ is given by $[[0, 1, -3], [-1, 0, 1], [3, -1, 0]]$. The proximity to the minimax strategy is measured by the value of $\max_{y \in \mathcal{Y}} (x^\mu)^\top A y - v^*$.

**Symmetric perturbation.** *Payoff perturbation* is an extensively studied technique for solving games (Facchinei & Pang, 2003; Liu et al., 2023). In this approach, the payoff functions of all players are perturbed by a strongly convex function $\psi$. For example, in bilinear games, instead of solving the original game in Eq. (1), we solve the following perturbed game:

$$\min_{x \in \mathcal{X}} \max_{y \in \mathcal{Y}} \left\{ x^\top A y + \mu \psi(x) - \mu \psi(y) \right\},$$

where $\mu \in (0, \infty)$ is the *perturbation strength*. Since the perturbation is applied to both players' payoff functions, we refer to this perturbed game as a *symmetrically* perturbed game. In this study, we specifically focus on the standard case, where the perturbation payoff function $\psi$ is given by the squared $\ell^2$-norm, i.e., $\psi(x) = \frac{1}{2}\|x\|^2$:

$$\min_{x \in \mathcal{X}} \max_{y \in \mathcal{Y}} \left\{ x^\top A y + \frac{\mu}{2} \|x\|^2 - \frac{\mu}{2} \|y\|^2 \right\}. \tag{2}$$

Let $x^\mu$ (resp. $y^\mu$) denote the minimax (resp. maximin) strategy in the symmetrically perturbed game Eq. (2) [2], which can be solved at an exponentially fast rate (Cen et al., 2021; 2023; Pattathil et al., 2023; Sokota et al., 2023). It is known that the solution $(x^\mu, y^\mu)$ is only an approximation of an equilibrium of the original game, with an error bounded by $\mathcal{O}(\mu)$ (Liu et al., 2023; Abe et al., 2024). Consequently, typical perturbation-based methods must employ a decreasing schedule for $\mu$, or use an extremely small fixed $\mu$ tuned to the number of iterations $T$ (e.g., $\mu = \mathcal{O}(1/T)$), which requires careful hyperparameter tuning (Tatarenko & Kamgarpour, 2019; Bernasconi et al., 2022; Cai et al., 2023). See Figure 1a for a biased Rock–Paper–Scissors game where the perturbed solution differs from the original equilibrium. We provide a rigorous theoretical justification for this behavior in Appendix B.

## 3 ASYMMETRIC PAYOFF PERTURBATION

In this section, we explain our novel technique of asymmetric payoff perturbation. We demonstrate that a seemingly minor structural change—perturbing only one player's payoff—can yield a dramatically different outcome: the solution of the perturbed game exactly matches an equilibrium strategy $x^*$ in many cases.

### 3.1 ASYMMETRIC PAYOFF PERTURBATION

Instead of incorporating the perturbation into both players' payoff functions, we consider the case where only player $x$'s payoff function is perturbed:

$$\min_{x \in \mathcal{X}} \max_{y \in \mathcal{Y}} \left\{ x^\top A y + \frac{\mu}{2} \|x\|^2 \right\}. \tag{3}$$

---

[2]The minimax strategy is uniquely determined because symmetrically perturbed games satisfy the strongly convex–strongly concave property.

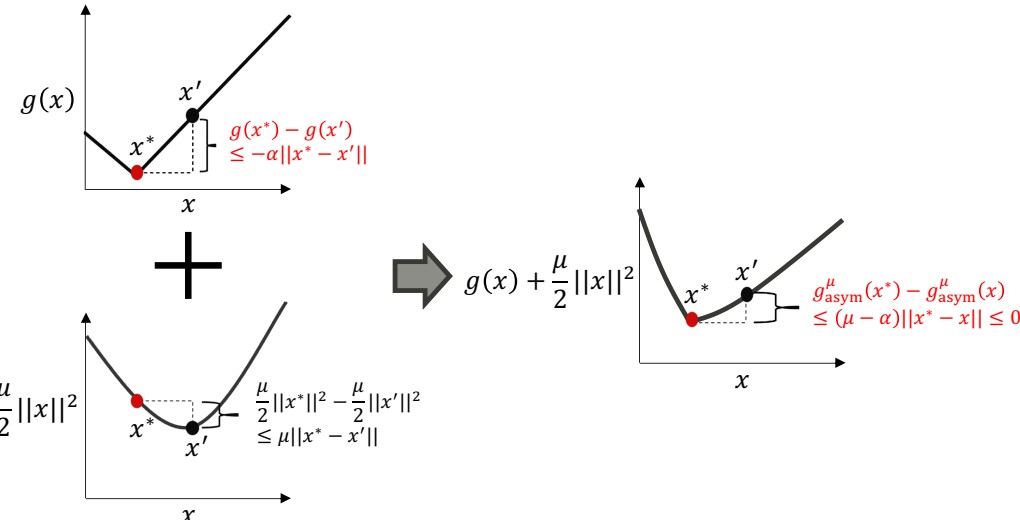

Figure 2: The landscape of the objective function for player $x$ in asymmetrically perturbed games. The functions $g(x)$ and $g_{\mathrm{asym}}^{\mu}(x)$ are defined as $g(x) := \max\limits_{y \in \mathcal{Y}} x^{\top} A y$ and $g_{\mathrm{asym}}^{\mu}(x) := g(x) + \frac{\mu}{2}\|x\|^2$, respectively.

The procedure we are going to describe in Theorem 3.1 focuses on computing the minimax strategy $x^*$, rather than the maximin strategy $y^*$. To compute $y^*$, we simply solve the corresponding maximin problem for player $y$:

$$\max_{y \in \mathcal{Y}} \min_{x \in \mathcal{X}} \left\{ x^{\top} A y - \frac{\mu}{2}\|y\|^2 \right\}.$$

The same reasoning applies to this perturbed maximin optimization problem. Thus, hereafter, we primarily focus on the perturbed game Eq. (3) from the perspective of player $x$.

Since the function $\max_{y \in \mathcal{Y}} x^{\top} A y$ is convex with respect to $x$ (Boyd & Vandenberghe, 2004), the perturbed objective $\max_{y \in \mathcal{Y}} x^{\top} A y + \frac{\mu}{2}\|x\|^2$ is $\mu$-strongly convex. Therefore, the minimax strategy for the perturbed game Eq. (3) is unique. We denote it by $x^{\mu}$ and denote the set of maximin strategies in Eq. (3) by $\mathcal{Y}^{\mu}$. Since both the minimax and maximin strategies constitute a Nash equilibrium of the perturbed game, the pair $(x^{\mu}, y^{\mu})$ with $y^{\mu} \in \mathcal{Y}^{\mu}$ satisfies the following conditions: for all $\tilde{y}^{\mu} \in \mathcal{Y}^{\mu}$ and $x \in \mathcal{X}$,

$$(x^{\mu})^{\top} A \tilde{y}^{\mu} + \frac{\mu}{2}\|x^{\mu}\|^2 \leq x^{\top} A \tilde{y}^{\mu} + \frac{\mu}{2}\|x\|^2, \tag{4}$$

and for all $y \in \mathcal{Y}$,

$$(x^{\mu})^{\top} A y^{\mu} \geq (x^{\mu})^{\top} A y. \tag{5}$$

### 3.2 EQUILIBRIUM INVARIANCE UNDER THE ASYMMETRIC PERTURBATION

In this section, we discuss the properties of the minimax strategies for asymmetrically perturbed games. Surprisingly, we can show that $x^{\mu}$ in Eq. (3) does correspond to a minimax strategy in the original game Eq. (1) for all $\mu$ smaller than a certain positive constant:

**Theorem 3.1.** *Assume that the perturbation strength $\mu$ is set such that $\mu \in (0, \frac{\alpha}{\max_{x \in \mathcal{X}}\|x\|})$, where $\alpha > 0$ is a constant depending only the game instance. Then, the minimax strategy $x^{\mu}$ in the corresponding asymmetrically perturbed game Eq. (3) satisfies $x^{\mu} \in \mathcal{X}^*$.*

Thus, whenever $\mu$ is below a certain positive constant, $x^{\mu}$ coincides with the minimax strategy of the original game. Figure 1b illustrates this feature in a simple example (in that example, invariance holds for $\mu < 2.5$).

**Remark 3.2** (Limitation of small fixed $\mu$)**.** The invariance result above holds in the small-$\mu$ regime: specifically, it requires $\mu \in \left(0, \alpha / \max_{x \in \mathcal{X}}\|x\|\right)$, where $\alpha > 0$ is a problem-dependent constant

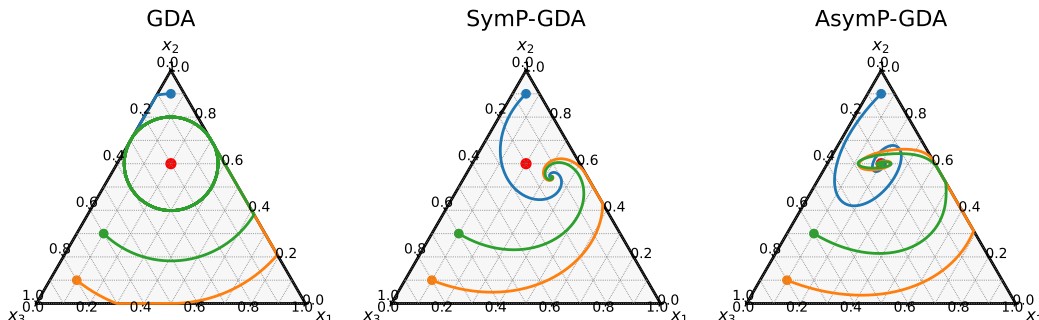

Figure 3: Trajectories of strategies for player $x$ using AsymP-GDA, SymP-GDA, and GDA. The game matrix $A$ is set to $A = [[0, 1, -3], [-1, 0, 1], [3, -1, 0]]$, and the strategy spaces are set to $\mathcal{X} = \mathcal{Y} = \Delta^3$. The red point represents the minimax strategy in the original game. The trajectories originate from different initial strategies, demonstrating the learning dynamics under each method.

determined by the geometry of $\mathcal{X}^*$. In instances with a small $\alpha$, the allowable $\mu$ can be quite small. Empirically, we did not observe such cases in our experiments (see Figures 1b and 4). Moreover, this sensitivity can be mitigated in practice by introducing an adaptive perturbation (Perolat et al., 2021; Abe et al., 2024). Note that the scheme is different from arbitrarily scheduling the perturbation strength mentioned in Section 1 and does not require careful hyperparameter tuning; see Remark 4.3 and Appendix D.

The key ingredient in proving Theorem 3.1 is the near-linear behavior of the objective function $g(x) := \max_{y \in \mathcal{Y}} x^\top A y$ for player $x$ in the original game. Specifically, according to Wei et al. (2021), there exists a constant $\alpha > 0$ such that:

$$\forall x^* \in \mathcal{X}^*, \ g(x) - g(x^*) \geq \alpha \|x - \Pi_{\mathcal{X}^*}(x)\|,$$

where $\Pi_{\mathcal{A}}(a) = \underset{a' \in \mathcal{A}}{\arg \min} \|a - a'\|$ represents the projection operator onto a given closed convex set $\mathcal{A}$. This inequality implies that deviating from the minimax strategy set $\mathcal{X}^*$ results in an increase in the objective function proportionally to the distance $\|x - \Pi_{\mathcal{X}^*}(x)\|$. In contrast, the variation (i.e., the gradient) of the perturbation payoff function $\frac{\mu}{2} \|x\|^2$ can always be bounded by $\mathcal{O}(\mu)$ over $\mathcal{X}$. Hence, by choosing $\mu$ sufficiently small, we can ensure that the perturbation payoff function does not significantly incentivize player $x$ to deviate from $x^*$. In Figure 2, we illustrate this fact intuitively. The addition of the strongly convex function $\frac{\mu}{2} \|x\|^2$ does not shift the optimum $x^*$ of the original $g(x)$ if $\mu$ is sufficiently small, as the kink of the lines through $g(x)$ dominates. The detailed proof of Theorem 3.1 is provided in Appendix F.

In summary, to compute a minimax strategy $x^* \in \mathcal{X}^*$ in the original game, it is sufficient to solve the asymmetrically perturbed game Eq. (3) with a small perturbation strength $\mu > 0$. Note that, as mentioned above, one can also compute a maximin strategy $y^* \in \mathcal{Y}^*$ by solving the game $\max_{y \in \mathcal{Y}} \min_{x \in \mathcal{X}} \left\{ x^\top A y - \frac{\mu}{2} \|y\|^2 \right\}$, where the payoff perturbation is applied only to player $y$.

## 4 ASYMMETRICALLY PERTURBED GRADIENT DESCENT ASCENT

This section proposes a first-order method, Asymmetrically Perturbed Gradient Descent Ascent (AsymP-GDA), for solving asymmetrically perturbed games Eq. (3). At each iteration $t \in [T]$, AsymP-GDA updates each player's strategy according to the following alternating updates[3]:

$$\begin{aligned} x^{t+1} &= \Pi_{\mathcal{X}} \left( x^t - \eta \left( A y^t + \mu x^t \right) \right), \\ y^{t+1} &= \Pi_{\mathcal{Y}} \left( y^t + \eta A^\top x^{t+1} \right), \end{aligned} \tag{6}$$

---

[3]AsymP-GDA employs alternating updates rather than simultaneous updates, as recent work has demonstrated the advantages of the former over the latter (Lee et al., 2024).

where $\eta > 0$ is the learning rate. In AsymP-GDA, player $x$'s strategy $x^t$ is updated based on the gradient of the perturbed payoff function $x^\top Ay + \frac{\mu}{2}\|x\|^2$, while player $y$'s strategy $y^t$ is updated using the gradient of the original payoff function $x^\top Ay$. AsymP-GDA adds only negligible per-iteration runtime or memory overhead relative to standard alternating GDA, with the only additional operation being a single vector addition.

Since the perturbed payoff function of player $x$ is strongly convex, it is anticipated that AsymP-GDA enjoys a last-iterate convergence guarantee. By combining this observation with Theorem 3.1, when $\mu$ is sufficiently small, the updated strategy $x^t$ should converge to a minimax strategy $x^*$ in the original game. We confirm this empirically by plotting the trajectory of $x^t$ updated by AsymP-GDA in a sample normal-form game, as shown in Figure 3. We also provide the trajectories of GDA and SymP-GDA; in the latter, the squared $\ell^2$-norm perturbs both players' gradients. For both AsymP-GDA and SymP-GDA, the perturbation strength is set to $\mu = 1$. As expected, AsymP-GDA successfully converges to the minimax strategy (red point) in the original game, whereas SymP-GDA converges to a point far from the minimax strategy, and GDA cycles around the minimax strategy. Further details and additional experiments in normal-form games can be found in Appendix A.3.

## 4.1 LAST-ITERATE CONVERGENCE RATE

In this section, we provide the convergence result of AsymP-GDA. Let $\|A\|$ denote the largest singular value of a given matrix $A$, and $D := \max_{z,z' \in \mathcal{X} \times \mathcal{Y}} \|z - z'\|$ denote the diameter of $\mathcal{X} \times \mathcal{Y}$. Theorem 4.1 demonstrates that AsymP-GDA converges to the minimax/maximin strategies in the asymmetrically perturbed game at a rate of $\mathcal{O}(1/t)$:

**Theorem 4.1.** *For an arbitrary perturbation strength $\mu > 0$, if the learning rate satisfies $\eta <$ $\min\left(\frac{\mu}{2(\mu^2+\|A\|^2)}, \frac{8(\mu+\|A\|)}{D\min\left(\mu,\frac{\beta}{\mu}\right)}\right)$, then $x^t$ and $y^t$ satisfy for any $t \geq 1$:*

$$\left\|x^\mu - x^t\right\|^2 + \left\|\Pi_{\mathcal{Y}^\mu}(y^t) - y^t\right\|^2 \leq \frac{256D^2(\mu + \|A\|)^2}{\eta^2 \min\left(\mu^2, \frac{\beta^2}{\mu^2}\right)t},$$

*where $\beta > 0$ is a positive constant depending only on $\mathcal{Y}^\mu$.*

Note that the statement of Theorem 4.1 holds for any fixed $\mu > 0$. Thus, by combining Theorems 3.1 and 4.1, we can conclude that if $\mu$ is sufficiently small (which does not need to depend on the number of iterations $t$ or $T$), player $x$'s strategy $x^t$ updated by AsymP-GDA converges to a minimax strategy in the original game Eq. (1):

**Corollary 4.2.** *Assume that the perturbation strength $\mu$ is set such that $\mu \in (0, \frac{\alpha}{\max_{x \in \mathcal{X}} \|x\|})$, and the learning rate is set such that $\eta < \min\left(\frac{\mu}{2(\mu^2+\|A\|^2)}, \frac{8(\mu+\|A\|)}{D\min\left(\mu,\frac{\beta}{\mu}\right)}\right)$. Then, AsymP-GDA ensures the convergence of $x^t$ to an equilibrium $x^*$ in the original game at a rate of $\mathcal{O}(1/t)$:*

$$\left\|x^* - x^t\right\|^2 \leq \frac{256D^2(\mu + \|A\|)^2}{\eta^2 \min\left(\mu^2, \frac{\beta^2}{\mu^2}\right)t}.$$

Corollary 4.2 provides an $\mathcal{O}(1/t)$ last-iterate convergence rate for AsymP-GDA. By contrast, optimistic methods such as OGDA and OMWU are known to achieve linear last-iterate convergence in certain settings (e.g., bilinear games) (Wei et al., 2021). Accordingly, our rate can be weaker in those regimes. Empirically, however, AsymP-GDA exhibits faster convergence in our experiments (see Figure 6 in Appendix A). We expect that a sharper analysis will yield a linear last-iterate convergence rate for AsymP-GDA. Establishing such a rate is a promising direction for future work. Furthermore, Corollary 4.2 shows that the constant in the $\mathcal{O}(1/t)$ scales as $\mathcal{O}(1/\mu^2)$. This dependency aligns with the trade-off between accuracy and convergence speed observed in Figure 8 in Appendix A.

**Remark 4.3** ($\mu$ can be chosen freely with an anchoring mechanism)**.** Recent perturbation-based methods for equilibrium learning (Perolat et al., 2021; Abe et al., 2024) adopt an *anchoring* mechanism and demonstrate strong empirical performance, together with theoretical support. In the anchoring mechanism, the perturbation term $\frac{\mu}{2}\|x\|^2$ is replaced with $\frac{\mu}{2}\|x - \sigma\|^2$, where the anchor point $\sigma$ is periodically updated to the current strategy $x^t$. AsymP-GDA can incorporate this mechanism; doing so

mitigates the small-$\mu$ requirement discussed above while preserving an $O(1/t)$ last-iterate guarantee of the same form as Theorem 4.1. We provide the formal statement and proof in Appendix D.

## 4.2 PROOF SKETCH OF THEOREM 4.1

This section outlines the proof sketch for Theorem 4.1. The complete proofs are provided in Appendix G.

**(1) Monotonic decrease of the distance function.** Firstly, leveraging the strong convexity of the perturbation payoff function, $\frac{\mu}{2} \|x\|^2$, we can show that the distance between the current strategy profile $z^t = (x^t, y^t)$ and any equilibrium $z^\mu = (x^\mu, y^\mu)$ monotonically decreases. Specifically, we have for any $t \geq 1$:

$$\left\|z^\mu - z^{t+1}\right\|^2 - \left\|z^\mu - z^t\right\|^2 \leq -\eta\mu \left\|x^\mu - x^{t+1}\right\|^2 - \frac{1}{2} \left\|z^{t+1} - z^t\right\|^2. \tag{7}$$

**(2) Lower bound on the path length.** The primary technical challenge is deriving the term related to the distance between $y^{t+1}$ and the maximin strategies set $\mathcal{Y}^\mu$, i.e., $\left\|\Pi_{\mathcal{Y}^\mu}(y^{t+1}) - y^{t+1}\right\|^2$, which leads to the last-iterate convergence rate. To this end, we first derive the following lower bound on the path length $\left\|z^{t+1} - z^t\right\|$ by the distance $\left\|x^\mu - x^{t+1}\right\|^2$ and $\left\|A(y^\mu - y^{t+1})\right\|^2$ (as shown in Lemmas G.3 and G.4):

$$\left\|z^{t+1} - z^t\right\| \geq \Omega\left(\left\|x^\mu - x^{t+1}\right\|^2 + \left\|A(y^\mu - y^{t+1})\right\|^2\right). \tag{8}$$

One might think that the term of $\left\|A(y^\mu - y^{t+1})\right\|^2$ can be 0 even if $y^{t+1} \notin \mathcal{Y}^\mu$. However, by proving that $Ay^*$ attains a unique vector $b^* \in \mathbb{R}^m$ for all $y^* \in \mathcal{Y}^\mu$, and this vector cannot be achieved by any strategy $y \notin \mathcal{Y}^\mu$, we demonstrate that $\left\|A(y^\mu - y^{t+1})\right\|^2 \neq 0$ as long as $y \notin \mathcal{Y}^\mu$. Using this fact, we demonstrate that the term of $\left\|A(y^\mu - y^{t+1})\right\|^2$ can be lower bounded by the distance between $y^{t+1}$ and the maximin strategy set:

$$\left\|A(y^\mu - y^{t+1})\right\|^2 \geq \Omega\left(\left\|\Pi_{\mathcal{Y}^\mu}(y^{t+1}) - y^{t+1}\right\|^2\right). \tag{9}$$

By combining Eq. (8) and Eq. (9), we obtain the following lower bound on $\left\|z^{t+1} - z^t\right\|$ by the distance between the current strategy profile and the equilibrium set:

$$\left\|z^{t+1} - z^t\right\| \geq \Omega\left(\left\|x^\mu - x^{t+1}\right\|^2 + \left\|\Pi_{\mathcal{Y}^\mu}(y^{t+1}) - y^{t+1}\right\|^2\right). \tag{10}$$

**(3) Last-iterate convergence rate.** Putting Eq. (10) into Eq. (7), we have for any $t \geq 1$:

$$\left\|\Pi_{\mathcal{Z}^\mu}(z^{t+1}) - z^{t+1}\right\|^2 \leq \left\|\Pi_{\mathcal{Z}^\mu}(z^t) - z^t\right\|^2 - \Omega\left(\left\|\Pi_{\mathcal{Z}^\mu}(z^{t+1}) - z^{t+1}\right\|^4\right),$$

where $\mathcal{Z}^\mu := \{x^\mu\} \times \mathcal{Y}^\mu$. Finally, utilizing an auxiliary lemma on recursive formulas by (Wei et al., 2021), we obtain the following upper bound on the distance $\left\|\Pi_{\mathcal{Z}^\mu}(z^t) - z^t\right\|^2$:

$$\left\|\Pi_{\mathcal{Z}^\mu}(z^t) - z^t\right\|^2 \leq \mathcal{O}(1/t).$$

$\square$

**Remark 4.4** (Technical challenge in proving Theorem 4.1)**.** The main technical challenge in proving Theorem 4.1 arises from the asymmetric nature of the perturbation, which is applied only to player $x$. Unlike the symmetric case, where strong convexity in both players' payoff functions directly yields contraction, the asymmetric setting requires a more subtle analysis since player $y$'s payoff function remains linear. Our proof establishes a last-iterate convergence rate by lower bounding the projection error of $y^t$ through the gap in payoff vectors $Ay^t - Ay^\mu$, as shown in Eq. (9), rather than relying on strong convexity. This contrasts with techniques for unconstrained linear-quadratic games (Zhang et al., 2022), where the dynamics reduce to a linear system; in our constrained setting, projections alter the dynamics fundamentally. See Appendix G for details.

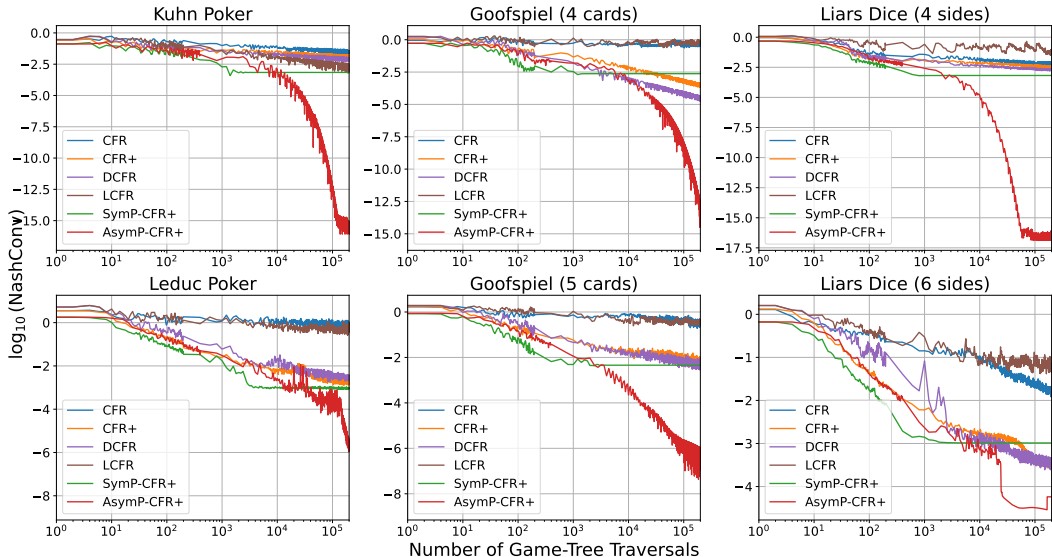

Figure 4: Performance in extensive-form games. Compared to the baselines, AsymP-CFR+ performs twice as many game-tree traversals per iteration to compute both players' equilibrium strategies. For a fair comparison, the x-axis reports the cumulative number of traversals: four per iteration for AsymP-CFR+ and two per iteration for the baselines.

## 5 ASYMMETRICALLY PERTURBED CFR+

We introduce the concept of asymmetric perturbation into learning algorithms for extensive-form games, which involve sequential decision-making and imperfect information, making them a more complex and realistic setting for strategic interactions. Here, we provide only empirical simulations, unlike in Section 4, as analyzing extensive-form games is significantly more difficult than analyzing normal-form games.

Extensive-form games are commonly solved using Counterfactual Regret Minimization (CFR) (Zinkevich et al., 2007) and its variants, such as CFR+ (Tammelin, 2014). In CFR+, the cumulative counterfactual regret $R_x(I, a)$ of player $x$ is updated to ensure it does not fall below zero:

$$R_x^{t+1}(I, a) = \max\left(R_x^t(I, a) + r_x^t(I, a), 0\right),$$

where $r_x^t(I, a)$ denotes the immediate counterfactual regret for information set $I$ and action $a$ at iteration $t$, computed from the original payoff function $u_x$. To investigate the impact of perturbations in extensive-form games, we propose Asymmetrically Perturbed CFR+ (AsymP-CFR+), and for comparison, we also evaluate a symmetrically perturbed counterpart (SymP-CFR+) that we use as a baseline. In AsymP-CFR+, we perturb only player $x$. Let the perturbed payoff at history $h$ be

$$u_x^{\mu,t}(h, a) = u_x(h, a) - \mu x^t(a|I(h)),$$

where $I(h)$ is the information set containing $h$, and $x^t(a|I(h))$ is player $x$'s probability of choosing action $a$ at that information set. Using $u_x^{\mu,t}$, we compute the corresponding immediate counterfactual regret in the standard way and denote it by $r_x^{\mu,t}(I, a)$. The cumulative counterfactual regret for player $x$ is then updated as:

$$R_x^{t+1}(I, a) = \max\left(R_x^t(I, a) + r_x^{\mu,t}(I, a), 0\right).$$

In contrast, SymP-CFR+ applies perturbation symmetrically to both players. See Appendix A.4 for the notations for extensive-form games and the pseudocode of AsymP-CFR+ (Algorithm 1). As with AsymP-GDA, AsymP-CFR+ adds only negligible per-iteration runtime or memory overhead relative to CFR+.

We compare the NashConv of the last-iterate strategy for AsymP-CFR+ against SymP-CFR+ and baseline algorithms, including CFR, CFR+, LCFR, and DCFR (both from Brown & Sandholm

(2019)). Our experiments focus on six different extensive-form games: Kuhn Poker, Leduc Poker, Goofspiel (with four-card and five-card variants), and Liar's Dice (with four-sided and six-sided dice), all of which are implemented using OpenSpiel (Lanctot et al., 2019). For both AsymP-CFR+ and SymP-CFR+, we set $\mu = 0.01$. For DCFR, we use OpenSpiel's default parameters.

Figure 4 shows the NashConv values for each game. As indicated by these results, AsymP-CFR+ not only converges faster than any other method in all games but also directly reaches an equilibrium strategy, whereas SymP-CFR+ converges near the equilibrium. These results confirm that the asymmetric perturbation leads to convergence in extensive-form games.

## 6 RELATED LITERATURE

Saddle-point optimization problems have attracted significant attention due to their applications in machine learning, such as training generative adversarial networks (Daskalakis et al., 2018). No-regret learning algorithms have been extensively studied with the aim of achieving either average-iterate or last-iterate convergence. To attain last-iterate convergence, many recent algorithms incorporate optimism (Rakhlin & Sridharan, 2013a;b), including optimistic multiplicative weights update (Daskalakis & Panageas, 2019; Lei et al., 2021b; Wei et al., 2021), optimistic gradient descent ascent (Daskalakis et al., 2018; Mertikopoulos et al., 2019; de Montbrun & Renault, 2022), and extra-gradient methods (Golowich et al., 2020; Mokhtari et al., 2020).

As an alternative approach, payoff perturbation has gained renewed attention. In this approach, players' payoff functions are regularized with strongly convex terms (Cen et al., 2021; 2023; Pattathil et al., 2023), which stabilizes the dynamics and leads to convergence. Some existing works have shown convergence to an approximate equilibrium under fixed perturbation (Sokota et al., 2023; Tuyls et al., 2006; Coucheney et al., 2015; Leslie & Collins, 2005; Abe et al., 2022; Hussain et al., 2023). To recover equilibria of the original game, later studies have employed a decreasing schedule or iterative regularization (Facchinei & Pang, 2003; Koshal et al., 2013; Yousefian et al., 2017; Bernasconi et al., 2022; Liu et al., 2023; Cai et al., 2023), or have updated the regularization center periodically (Perolat et al., 2021; Abe et al., 2023; 2024). In contrast to these approaches, our algorithms converge without decaying or modifying the perturbation.

Extensive-form games, which model sequential and imperfect-information interactions, have also been studied from both theoretical and empirical perspectives. Strategy representations can be broadly classified into sequence-form (von Stengel, 1996) and behavioral-form. Under the sequence-form, optimistic algorithms have been shown to enjoy last-iterate convergence guarantees (Lee et al., 2021). Under the behavioral-form, perturbation-based approaches have also been applied successfully (Perolat et al., 2021; Sokota et al., 2023; Liu et al., 2023). Our asymmetric perturbation is compatible with both representations. Notably, our theoretical results in Theorem 4.1 and Corollary 4.2 hold under the sequence-form, and we empirically demonstrate strong performance under the behavioral-form too.

## 7 CONCLUSION AND LIMITATIONS

This paper introduces an asymmetric perturbation technique for solving saddle-point optimization problems, addressing key challenges in learning dynamics and equilibrium computation. Unlike symmetric perturbation methods that yield only approximate equilibria, our approach guarantees last-iterate convergence to the exact equilibrium without requiring parameter adjustment. This structural insight suggests new algorithmic designs that further exploit the asymmetric perturbation.

Our theoretical results target bilinear two-player zero-sum games. The key insight of equilibrium invariance in Theorem 3.1 relies on the near-linear growth of the objective function. We believe that an analogous formulation can be posed beyond bilinear games, including two-player zero-sum Markov games.

The invariance result in Theorem 3.1 holds in the small-$\mu$ regime and depends on a problem-dependent constant $\alpha > 0$ (Remark 3.2). When $\alpha$ is small, the admissible $\mu$ may be very small. We conjecture that, in such cases, using a larger $\mu$ does not drive the solution far from an equilibrium strategy of the original game. Finally, the provided rate of $\mathcal{O}(1/t)$ in Corollary 4.2 can be slower than the rates known for optimistic or decaying symmetric perturbation methods (Wei et al., 2021; Liu et al., 2023). We expect that a sharper analysis can yield a linear rate for AsymP-GDA.

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

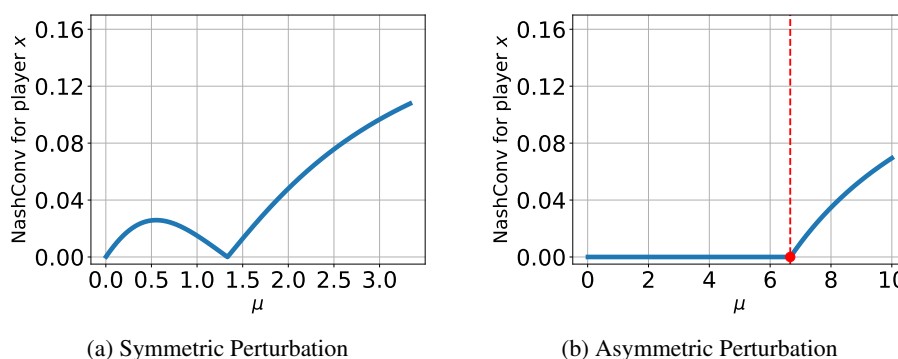

(a) Symmetric Perturbation          (b) Asymmetric Perturbation

Figure 5: The proximity of $x^\mu$ to $x^*$ under the symmetric perturbation and the asymmetric perturbation with varying $\mu$. The game matrix $A$ is given by $[[\frac{1}{3}, -\frac{2}{3}], [-\frac{2}{3}, 1]]$.

# A  EXPERIMENTAL DETAILS AND ADDITIONAL EXPERIMENTAL RESULTS

## A.1  INFORMATION ON THE COMPUTER RESOURCES

All experiments in this paper were conducted on macOS Sonoma 14.4.1 with Apple M2 Max and 32GB RAM.

## A.2  PROXIMITY TO EQUILIBRIUM UNDER SYMMETRIC AND ASYMMETRIC PERTURBATIONS IN BIASED MATCHING PENNIES

This section investigates the proximity of $x^\mu$ to the equilibrium $x^*$ in the Biased Matching Pennies (BMP) game under the symmetric/asymmetric payoff perturbation, with varying perturbation strength $\mu$. The game matrix for BMP is provided in Table 1.

Table 1: Game matrix in BMP

|       | $y_1$ | $y_2$ |
|-------|-------|-------|
| $x_1$ | $1/3$ | $-2/3$ |
| $x_2$ | $-2/3$ | $1$ |

BMP has a unique equilibrium $x^* = y^* = (\frac{5}{8}, \frac{3}{8})$, and the game value is given as $v^* = -\frac{1}{24}$.

Figure 5 exhibits the proximity of $x^\mu$ to $x^*$ as $\mu$ varies. Notably, under the symmetric perturbation, $x^\mu$ coincides with $x^*$ when $\mu$ is set to $\mu = \frac{\left(\frac{\mathbf{1}_m}{m}\right)^\top A\left(\frac{\mathbf{1}_n}{n}\right) - v^*}{\|x^*\|^2 - \frac{1}{m}} = \frac{4}{3}$. This result underscores the statement in Theorem B.1, that $x^\mu$ does not coincide with $x^*$ as long as $\mu \neq \frac{\left(\frac{\mathbf{1}_m}{m}\right)^\top A\left(\frac{\mathbf{1}_n}{n}\right) - v^*}{\|x^*\|^2 - \frac{1}{m}}$.

## A.3  ADDITIONAL EXPERIMENTS IN NORMAL-FORM GAMES

In this section, we experimentally compare our AsymP-GDA with SymP-GDA, GDA, and OGDA (Daskalakis et al., 2018; Wei et al., 2021). We conduct experiments on two normal-form games: Biased Rock-Paper-Scissors (BRPS) and Multiple Nash Equilibria (M-Ne). These games are taken from Abe et al. (2023) and Wei et al. (2021). Tables 2 and 3 provide the game matrices for BRPS and M-Ne, respectively.

Figure 6 illustrates the logarithm of NashConv averaged over 100 different random seeds. For each random seed, the initial strategies $(x^0, y^0)$ are chosen uniformly at random within the strategy spaces $\mathcal{X} = \Delta^m$ and $\mathcal{Y} = \Delta^n$. We use a learning rate of $\eta = 0.01$ for each algorithm, and a perturbation strength of $\mu = 1$ for both AsymP-GDA and SymP-GDA. We observe that AsymP-GDA converges to the minimax strategy in the original game, while SymP-GDA converges to a point far from the minimax strategy.

Table 2: Game matrix in BRPS

|       | $y_1$ | $y_2$ | $y_3$ |
| ----- | ----- | ----- | ----- |
| $x_1$ | 0     | 1     | $-3$  |
| $x_2$ | $-1$  | 0     | 1     |
| $x_3$ | 3     | $-1$  | 0     |

Table 3: Game matrix in M-Ne

|       | $y_1$ | $y_2$ | $y_3$ | $y_4$ | $y_5$ |
| ----- | ----- | ----- | ----- | ----- | ----- |
| $x_1$ | 0     | $-1$  | 1     | 0     | 0     |
| $x_2$ | 1     | 0     | $-1$  | 0     | 0     |
| $x_3$ | $-1$  | 1     | 0     | 0     | 0     |
| $x_4$ | $-1$  | 1     | 0     | 2     | $-1$  |
| $x_5$ | $-1$  | 1     | 0     | $-1$  | 2     |

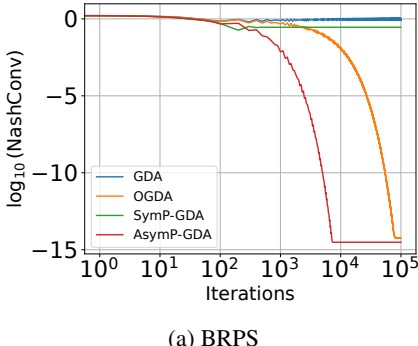

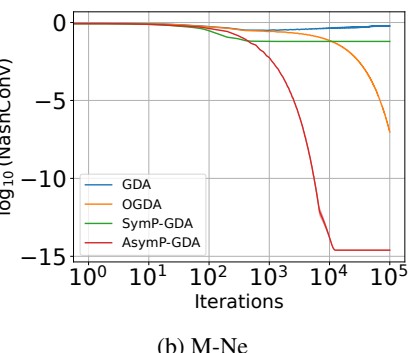

(a) BRPS

(b) M-Ne

Figure 6: Performance of AsymP-GDA, SymP-GDA, GDA, and OGDA in normal-form games. The shaded area represents the standard errors.

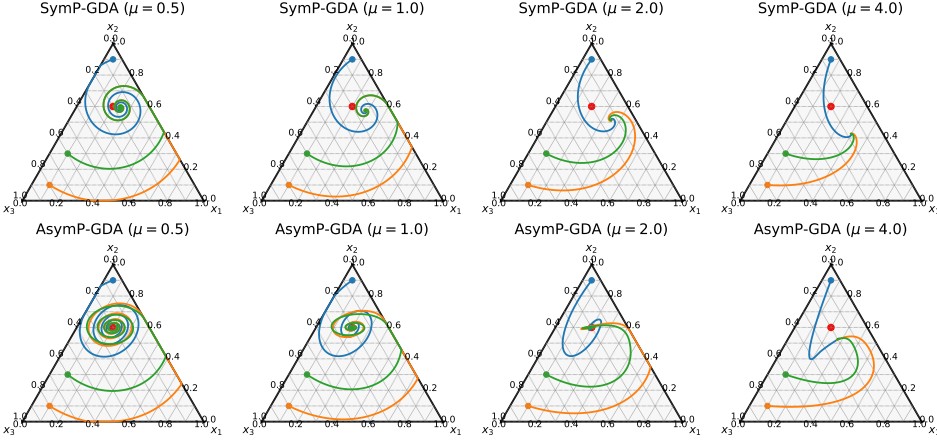

Figure 7: Trajectories for SymP-GDA (top row) and AsymP-GDA (bottom row) under different perturbation strengths $\mu \in \{0.5, 1.0, 4.0\}$ in BRPS.

Figure 7 illustrates the trajectories of SymP-GDA (top row) and AsymP-GDA (bottom row) under varying perturbation strengths $\mu \in \{0.5, 1.0, 2.0, 4.0\}$ in BRPS. For SymP-GDA, the trajectories do not converge directly to the equilibrium even for small values of $\mu = 0.5, 1.0$. Instead, they follow circuitous and elongated paths, resulting in slower convergence. Conversely, as $\mu$ increases ($\mu = 2.0, 4.0$), the trajectories become more direct, leading to faster convergence, but they remain farther from the equilibrium. In contrast, AsymP-GDA leads to direct convergence to the equilibrium with small perturbation strengths. For $\mu$ values up to $2.0$ the trajectories converge directly to the equilibrium. However, as $\mu$ increases beyond a threshold ($\mu = 4.0$), the trajectory deviates from the equilibrium. These results provide a more detailed understanding of the trends observed in Figures 1a and 1b, further illustrating the differences in convergence dynamics between symmetric and asymmetric perturbations.

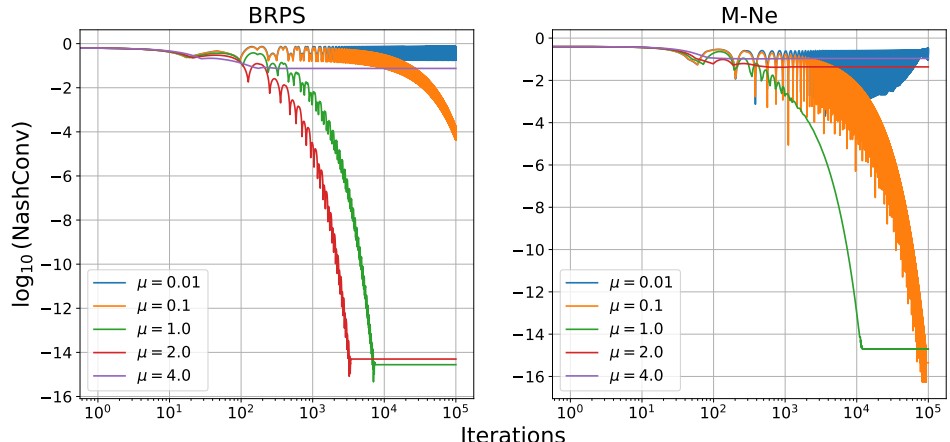

Figure 8: Sensitivity to the perturbation strength for AsymP-GDA with $\eta = 0.01$ in BRPS and M-Ne.

Figure 8 illustrates the performance of AsymP-GDA on BRPS and M-Ne with $\mu \in \{0.01, 0.1, 1.0, 2.0, 4.0\}$ and $\eta = 0.01$. For sufficiently small $\mu$, the limit point coincides with an equilibrium of the original game. However, decreasing $\mu$ also slows convergence. Overall, these results highlight a trade-off between accuracy and convergence speed.

### A.4 PSEUDO CODE FOR EXPERIMENTS IN EXTENSIVE-FORM GAMES

In this section, we present the pseudocode for AsymP-CFR+. First, we formally define a two-player extensive-form zero-sum game with imperfect information as a tuple $\langle N, c, H, Z, A, P, \pi_c, u, \mathcal{I} \rangle$. $N = \{x, y, c\}$ is a finite set of players and a chance player $c$. $H = \bigcup_{p \in N \cup \{c\}} H_p$ is the set of all possible *histories*, where each history is a sequence of *actions* and $H_p$ is the set of histories of player $p$'s action. $Z \subset H$ is the set of *terminal histories* where the game has ended and the player has no available actions. At each history $h \in H \setminus Z$, the current player chooses an action $a \in A(h)$. We denote $A(h)$ as the set of actions available at history $h$ that lead to a successor history $(ha) \in H$. A *player function* $P : H \setminus Z \to N \cup \{c\}$ maps each history $h$ to the player that chooses the next action at $h$. The chance player $c$ acts according to the defined distribution $\pi_c(\cdot|h) \in \Delta(A(h))$. A *payoff function* $u_x(h, a)$ (resp. $u_y(h, a)$) maps each history $h \in H$ and action $a \in A(h)$ to a real value for player $x$ (resp. player $y$).

For player $x$ (resp player $y$), the collection of *information sets* $\mathcal{I}_x \in \mathcal{I}$ are *information partitions* of the histories $\{h \in H|P(h) = x\}$. Player $x$ (resp player $y$) does not observe the true history $h$, but only the information set $I \in \mathcal{I}_x$ (resp player $y$) corresponding to $h$. This implies that for each information set $I$, if any two histories $h, h'$ belong to $I$, these histories are indistinguishable to the player: $A(h) = A(h')$ for any $h, h' \in I$, which we then denote $A(I)$. We also denote $I(h)$ as an information set containing history $h$.

Using these notations, the pseudocode for AsymP-CFR+ is provided in Algorithm 1.

## B IMPOSSIBILITY RESULTS FOR SYMMETRIC PERTURBATION

As we stated in Section 2, existing works (Liu et al., 2023; Abe et al., 2024) have shown that the distance between the solution of the symmetrically perturbed game $(x^\mu, y^\mu)$ and the solution in the original game $(x^*, y^*)$ is upper bounded by $\mathcal{O}(\mu)$. However, they do not guarantee that the two solutions coincide, even for a small $\mu > 0$. In contrast, the following theorem provides the first formal impossibility result, proving that $(x^\mu, y^\mu)$ almost never coincides with $(x^*, y^*)$.

**Theorem B.1.** *Consider a normal-form game with a unique interior equilibrium. Assume that at this equilibrium, neither player chooses their actions uniformly at random, i.e., $(x^*, y^*) \neq \left(\frac{1}{m}\mathbf{1}_m, \frac{1}{n}\mathbf{1}_n\right)$.*

*Then, for any $\mu > 0$ such that $\mu \neq \frac{\left(\frac{\mathbf{1}_m}{m}\right)^\top A\left(\frac{\mathbf{1}_n}{n}\right) - v^*}{\|x^*\|^2 - \frac{1}{m}}$, the minimax strategy $x^\mu$ in Eq. (2) satisfies*

---

**Algorithm 1:** AsymP-CFR+

1 $R_x[I,a] \leftarrow 0$ and $R_y[I,a] \leftarrow 0$ for all information set $I$ and action $a$.
2 Initialize both players' strategy $(x^1, y^1)$ as a uniform distribution for all $I$ and $a$.
3 **for** $t = 1, \cdots, T$ **do**
4     Compute counterfactual value for player $x$ by $(Q[I,a])_{I,a} \leftarrow$ VALUECOMPUTE$_x(x^t, y^t, \mu)$.
5     Compute immediate counterfactual regret for player $x$ by

$$r[I,a] \leftarrow Q[I,a] - \sum_b x(b|I)Q[I,b]$$

.
6     Update cumulative counterfactual regret for player $x$ by

$$R_x[I,a] \leftarrow \max\left(R_y[I,a] + r[I,a], 0\right)$$

.
7     Update player $x$'s strategy by $x^{t+1}(a|I) = \frac{R_x[I,a]}{\sum_b R_x[I,b]}$ for all $I$ and $a$.
8     Compute counterfactual value for player $y$ by $(Q[I,a])_{I,a} \leftarrow$ VALUECOMPUTE$_y(x^{t+1}, y^t)$.
9     Compute immediate counterfactual regret for player $y$ by

$$r[I,a] \leftarrow Q[I,a] - \sum_b x(b|I)Q[I,b]$$

.
10    Update cumulative counterfactual regret for player $y$ by

$$R_y[I,a] \leftarrow \max\left(R_y[I,a] + r[I,a], 0\right)$$

.
11    Update player $y$'s strategy by $y^{t+1}(a|I) = \frac{R_y[I,a]}{\sum_b R_y[I,b]}$ for all $I$ and $a$.
12 **end for**

---

**Algorithm 2:** VALUECOMPUTE$_x(x, y, \mu)$ for player $x$

1 $Q[I,a] \leftarrow 0$ for all information set $I$ and action $a$.
2 TRAVERSE$(\emptyset, 1)$
3 **return** $(Q[I,a])_{I,a}$

---

4 **subroutine** TRAVERSE $(h, \rho_{-i})$
5     **if** $h \in Z$ **then**
6         **return** 0
7     **else if** $P(h) = c$ **then**
8         **return** $\sum_{a \in A(h)} \pi_c(a|h) \cdot ($TRAVERSE $(ha, \pi_c(a|h) \cdot \rho_{-i}) + u_x(h,a))$
9     **else if** $P(h) = y$ **then**
10        **return** $\sum_{a \in A(h)} y(a|I(h)) \cdot ($TRAVERSE $(ha, y(a|I(h)) \cdot \rho_{-i}) + u_x(h,a))$
11    **end if**
12    Let $I$ be the information set containing $h$
13    **if** $P(h) = x$ **then**
14        $q[h] \leftarrow 0$
15        $q[h,a] \leftarrow 0$ for all $a \in A(h)$
16        **for** $a \in A(h)$ **do**
17            $q[h,a] \leftarrow$ TRAVERSE$(ha, \rho_{-i}) + u_x(h,a) - \mu x(a|I)$
18            $Q[I,a] \leftarrow Q[I,a] + \rho_{-i} \cdot q[h,a]$
19            $q[h] \leftarrow q[h] + x(a|I) \cdot q[h,a]$
20        **end for**
21        **return** $q[h]$

---

**Algorithm 3:** VALUECOMPUTE$_y(x, y)$ for player $y$

---

1   $Q[I, a] \leftarrow 0$ for all information set $I$ and action $a$
2   TRAVERSE$(\emptyset, 1)$
3   **return** $(Q[I, a])_{I,a}$

---

4   **subroutine** TRAVERSE $(h, \rho_{-i})$
5     **if** $h \in Z$ **then**
6       **return** $0$
7     **else if** $P(h) = c$ **then**
8       **return** $\sum_{a \in A(h)} \pi_c(a|h) \cdot (\text{TRAVERSE}\,(ha, \pi_c(a|h) \cdot \rho_{-i}) + u_y(h, a))$
9     **else if** $P(h) = x$ **then**
10      **return** $\sum_{a \in A(h)} x(a|I(h)) \cdot (\text{TRAVERSE}\,(ha, x(a|I(h)) \cdot \rho_{-i}) + u_y(h, a))$
11    **end if**
12    Let $I$ be the information set containing $h$
13    **if** $P(h) = y$ **then**
14      $q[h] \leftarrow 0$
15      $q[h, a] \leftarrow 0$ for all $a \in A(h)$
16      **for** $a \in A(h)$ **do**
17        $q[h, a] \leftarrow \text{TRAVERSE}(ha, \rho_{-i}) + u_y(h, a)$
18        $Q[I, a] \leftarrow Q[I, a] + \rho_{-i} \cdot q[h, a]$
19        $q[h] \leftarrow q[h] + y(a|I) \cdot q[h, a]$
20      **end for**
21      **return** $q[h]$

---

$x^\mu \neq x^*$. *Furthermore, for any $\mu > 0$ such that $\mu \neq \frac{v^* - \left(\frac{\mathbf{1}_m}{m}\right)^\top A\left(\frac{\mathbf{1}_n}{n}\right)}{\|y^*\|^2 - \frac{1}{n}}$, the maximin strategy $y^\mu$ in Eq. (2) satisfies $y^\mu \neq y^*$.*

The proof is provided in Appendix E. Additionally, we extend our analysis to the case where both players have different perturbation strengths, i.e., $\mu_x > 0$ and $\mu_y > 0$, as shown in Appendix C.

**Discussion on Theorem B.1.** The term $\frac{\left(\frac{\mathbf{1}_m}{m}\right)^\top A\left(\frac{\mathbf{1}_n}{n}\right) - v^*}{\|x^*\|^2 - \frac{1}{m}}$ can be interpreted as a measure of the difference between the equilibrium $(x^*, y^*)$ and the uniform random strategy profile $\left(\frac{1}{m}\mathbf{1}_m, \frac{1}{n}\mathbf{1}_n\right)$. Specifically, the numerator $\left(\frac{\mathbf{1}_m}{m}\right)^\top A\left(\frac{\mathbf{1}_n}{n}\right) - v^*$ represents the difference in the payoffs, while the denominator $\|x^*\|^2 - \frac{1}{m}$ represents the difference in the squared $\ell^2$-norms, respectively. A promising direction for future research is to theoretically demonstrate that, when $\mu = \frac{v^* - \left(\frac{\mathbf{1}_m}{m}\right)^\top A\left(\frac{\mathbf{1}_n}{n}\right)}{\|y^*\|^2 - \frac{1}{n}}$, the corresponding equilibrium coincides exactly with the equilibrium in the original game, i.e., $(x^\mu, y^\mu) = (x^*, y^*)$. We have experimentally confirmed this, and the results are presented in the Appendix A.2.

**When the game is symmetric.** Next, let us consider the case when $A^\top = -A$, as in Rock-Paper-Scissors. In this scenario, the equilibrium strategies $x^\mu$ and $y^\mu$ are not identical to the minimax or maximin strategies of the original game, regardless of the choice of $\mu > 0$.

**Corollary B.2.** *Assume that $A^\top = -A$. Under the same setup as Theorem B.1, the equilibrium $(x^\mu, y^\mu)$ in Eq. (2) always satisfies $x^\mu \neq x^*$ and $y^\mu \neq y^*$ for any $\mu > 0$.*

This is because it always holds that $v^* - \left(\frac{\mathbf{1}_m}{m}\right)^\top A\left(\frac{\mathbf{1}_n}{n}\right) = 0$ when $A^\top = -A$. Figure 1a shows the proximity of $x^\mu$ to $x^*$ with varying perturbation strength $\mu$ in a simple biased Rock-Paper-Scissors game. We observe that as long as $\mu > 0$, $x^\mu$ remains distant from $x^*$. This observation supports the theoretical results in Theorem B.1 and Corollary B.2.

# C INDEPENDENTLY PERTURBED GAME

Let us consider the perturbed game where players $x$ and $y$ choose independently their perturbation strengths $\mu_x$ and $\mu_y$:

$$\min_{x \in \mathcal{X}} \max_{y \in \mathcal{Y}} \left\{ x^\top A y + \frac{\mu_x}{2} \|x\|^2 - \frac{\mu_y}{2} \|y\|^2 \right\}. \tag{11}$$

We establish a theoretical result similar to Theorem 3.1 for this perturbed game.

**Theorem C.1.** *Assume that the original game is a normal-form game with a unique interior equilibrium, and that $(x^*, y^*) \neq \left( \frac{\mathbf{1}_m}{m}, \frac{\mathbf{1}_n}{n} \right)$, i.e., the equilibrium is not the uniform random strategy profile. Then, for any $\mu_x > 0$ such that $\mu \neq \frac{\left( \frac{\mathbf{1}_m}{m} \right)^\top A \left( \frac{\mathbf{1}_n}{n} \right) - v^*}{\|x^*\|^2 - \frac{1}{m}}$, the minimax strategy $x^\mu$ in the corresponding symmetrically perturbed game Eq. (11) satisfies $x^\mu \neq x^*$. Furthermore, for any $\mu > 0$ such that $\mu_y \neq \frac{v^* - \left( \frac{\mathbf{1}_m}{m} \right)^\top A \left( \frac{\mathbf{1}_n}{n} \right)}{\|y^*\|^2 - \frac{1}{n}}$, the maximin strategy $y^\mu$ in Eq. (11) satisfies $y^\mu \neq y^*$.*

# D ADAPTIVELY ASYMMETRIC PERTURBATION

As noted in Remark 3.2, AsymP-GDA may require a very small value of $\mu$ to ensure convergence to an equilibrium in the original game. To address this limitation, this section introduces an enhanced variant called Adaptively AsymP-GDA (Ada-AsymP-GDA), which incorporates an *adaptive anchoring strategy* $\sigma \in \mathcal{X}$ (Perolat et al., 2021; Abe et al., 2024).

In Ada-AsymP-GDA, instead of using the perturbation term $\mu x^t$ as in the original algorithm, we apply the gradient of the squared distance between $x^t$ and the anchor $\sigma$, namely $\mu(x^t - \sigma)$. The anchoring strategy $\sigma$ is updated periodically: it is reset to the current strategy $x^t$ every $T_\sigma$ iterations.

Let $k(t)$ denote the number of times $\sigma$ has been updated up to iteration $t$, and let $\sigma^{k(t)}$ denote the anchoring strategy after $k(t)$ updates. Since $\sigma$ is updated every $T_\sigma$ iterations, we have $k(t) = \lfloor (t-1)/T_\sigma \rfloor + 1$ and $\sigma^{k(t)} = x^{T_\sigma(k(t)-1)+1}$. In summary, Ada-AsymP-GDA updates each player's strategy at iteration $t \in [T]$ according to:

$$\begin{aligned} x^{t+1} &= \Pi_\mathcal{X} \left( x^t - \eta \left( A y^t + \mu(x^t - \sigma^{k(t)}) \right) \right), \\ y^{t+1} &= \Pi_\mathcal{Y} \left( y^t + \eta A^\top x^{t+1} \right). \end{aligned} \tag{12}$$

## D.1 LAST-ITERATE CONVERGENCE RATE

We now establish the last-iterate convergence rate of Ada-AsymP-GDA. In particular, we show that the final strategy $x^{T+1}$ converges to an equilibrium of the original game at a rate of $\mathcal{O}(1/T)$.

**Theorem D.1.** *Let $\mu > 0$ be an arbitrary perturbation strength. If we set $\eta < \min \left( \frac{\mu}{2(\mu^2 + \|A\|^2)}, \frac{8(\mu + \|A\|)}{D \min \left( \mu, \frac{\|A\|^2}{\mu} \right)} \right)$ and $T_\sigma = cT$ for some constant $c \leq \min \left( 1, \frac{\alpha^2}{2\mu^2 \|\Pi_{\mathcal{X}^*}(x^1) - x^1\|^2} \right)$, then the strategy $x^{T+1}$ satisfies:*

$$\left\| \Pi_{\mathcal{X}^*}(x^{T+1}) - x^{T+1} \right\|^2 \leq \mathcal{O} \left( \frac{1}{T} \right).$$

Notably, Ada-AsymP-GDA achieves a last-iterate convergence rate of $\mathcal{O}(1/T)$ even when using a relatively large value of $\mu$.

## D.2 EMPIRICAL PERFORMANCE IN NORMAL-FORM GAMES

In this section, we empirically evaluate the performance of Ada-AsymP-GDA on the same normal-form games presented in Appendix A.3. We compare Ada-AsymP-GDA against AsymP-GDA, SymP-GDA, and Ada-SymP-GDA, where Ada-SymP-GDA applies perturbations $\mu(x^t - \sigma_x^{k(t)})$ and $\mu(y^t - \sigma_y^{k(t)})$ to both players' gradients.

For all algorithms, we use a learning rate of $\eta = 0.01$ and a perturbation strength of $\mu = 5$. The update interval for the anchoring strategy is set to $T_\sigma = 10,000$ for both Ada-AsymP-GDA and Ada-SymP-GDA.

Figure 9 illustrates the logarithm of NashConv averaged over 100 random seeds. Ada-AsymP-GDA exhibits substantially faster convergence to an equilibrium compared to the other methods.

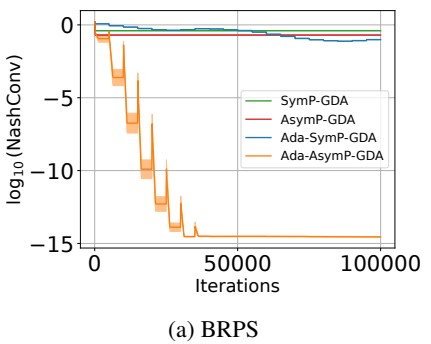
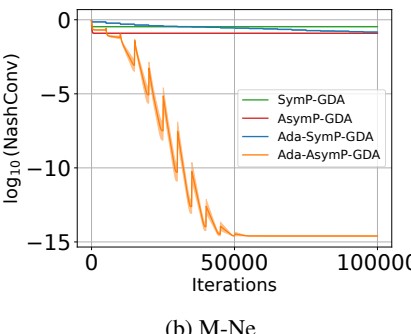

(a) BRPS

(b) M-Ne

Figure 9: Performance of Ada-AsymP-GDA in normal-form games. The shaded area represents the standard errors.

### D.3 EMPIRICAL PERFORMANCE IN EXTENSIVE-FORM GAMES

We also extend our method to extensive-form games through a variant called Adaptive AsymP-CFR+ (Ada-AsymP-CFR+), which integrates the adaptive anchoring strategy into AsymP-CFR+. Specifically, in Ada-AsymP-CFR+, the cumulative counterfactual regret $R_x^t$ for player $x$ under the following perturbed payoff function:

$$u_x^{\mu,t}(h,a) = u_x(h,a) - \mu\left(x^t(a|I(h)) - \sigma^{k(t)}(a|I(h))\right),$$

We compare Ada-AsymP-CFR+ with AsymP-CFR+, SymP-CFR+, and Ada-SymP-CFR+ in the same extensive-form games in Section 5. Note that Ada-SymP-CFR+ applies symmetric perturbations with the adaptive anchoring strategy. For all algorithms, we use a perturbation strength of $\mu = 0.05$. The update interval for the anchoring strategy is set to $T_\sigma = 2,500$ for both Ada-AsymP-CFR+ and Ada-SymP-CFR+.

Figure 10 illustrates the NashConv values for each game. The results indicate that Ada-AsymP-CFR+ consistently achieves lower NashConv values, demonstrating superior convergence performance.

## E PROOF OF THEOREM B.1

*Proof of Theorem B.1.* First, we prove that $y^\mu \neq y^*$ under the assumption that $\mu \neq \frac{v^* - \left(\frac{\mathbf{1}_m}{m}\right)^\top A\left(\frac{\mathbf{1}_n}{n}\right)}{\left(\|y^*\|^2 - \frac{1}{n}\right)}$ by contradiction. We assume that $y^\mu = y^*$. Since $(x^*, y^*)$ is in the interior of $\Delta^m \times \Delta^n$, we have:

$$\begin{aligned}
(A^\top x^*)_i &= v^*, \ \forall i \in [m] \\
(Ay^*)_i &= v^*, \ \forall i \in [n].
\end{aligned} \tag{13}$$

Then, from Eq. (13), we have for any $x \in \Delta^m$:

$$x^\top Ay^\mu + \frac{\mu}{2}\|x\|^2 = x^\top Ay^* + \frac{\mu}{2}\|x\|^2 = v^* + \frac{\mu}{2}\|x\|^2. \tag{14}$$

On the other hand,

$$\left(\frac{\mathbf{1}_m}{m}\right)^\top A^\top y^\mu + \frac{\mu}{2}\left\|\frac{\mathbf{1}_m}{m}\right\|^2 = \left(\frac{\mathbf{1}_m}{m}\right)^\top A^\top y^* + \frac{\mu}{2}\left\|\frac{\mathbf{1}_m}{m}\right\|^2 = v^* + \frac{\mu}{2}\left\|\frac{\mathbf{1}_m}{m}\right\|^2. \tag{15}$$

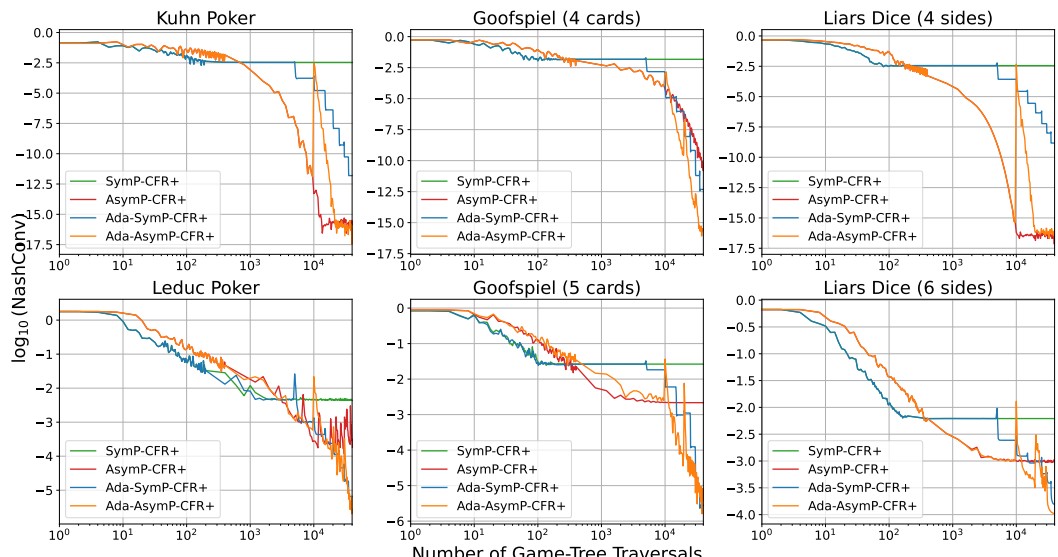

Figure 10: Performance of Ada-AsymP-CFR+ in extensive-form games.

By combining Eq. (14) and Eq. (15), we have for any $x \in \Delta^m$:

$$x^\top A y^\mu + \frac{\mu}{2} \|x\|^2 \geq \left( \frac{\mathbf{1}_m}{m} \right)^\top A^\top y^\mu + \frac{\mu}{2} \left\| \frac{\mathbf{1}_m}{m} \right\|^2.$$

Hence, from the property of the player $x$'s equilibrium strategy in the perturbed game, $x^\mu$ must satisfy $x^\mu = \frac{\mathbf{1}_m}{m}$.

On the other hand, from the property of the player $y$'s equilibrium strategy $y^\mu$ in the perturbed game, $y^\mu$ is an optimal solution of the following optimization problem:

$$\max_{y \in \Delta^n} \left\{ (x^\mu)^\top A y - \frac{\mu}{2} \|y\|^2 \right\}.$$

Let us define the following Lagrangian function $L(y, \kappa, \lambda)$ as:

$$L(y, \kappa, \lambda) = (x^\mu)^\top A y - \frac{\mu}{2} \|y\|^2 - \sum_{i=1}^n \kappa_i g_i(y) - \lambda h(y),$$

where $g_i(y) = -y_i$ and $h(y) = \sum_{i=1}^n y_i - 1$. Then, from the KKT conditions, we get the stationarity:

$$A^\top x^\mu - \mu y^\mu - \sum_{i=1}^n \kappa_i \nabla g_i(y^\mu) - \lambda \nabla h(y^\mu) = \mathbf{0}_n, \tag{16}$$

and the complementary slackness:

$$\forall i \in [n], \ \kappa_i g_i(y^\mu) = 0. \tag{17}$$

Since $y^\mu = y^*$ and $y^*$ is in the interior of $\Delta^n$, we have $g(y^\mu) = -y_i^\mu < 0$ for all $i \in [n]$. Thus, from Eq. (17), we have $\kappa_i = 0$ for all $i \in [n]$. Substituting this into Eq. (16), we obtain:

$$A^\top x^\mu - \mu y^\mu - \lambda \nabla h(y^\mu) = A^\top x^\mu - \mu y^\mu - \lambda \mathbf{1}_n = \mathbf{0}_n. \tag{18}$$

Hence, we have:

$$\lambda = \frac{\mathbf{1}_n^\top A^\top x^\mu - \mu}{n}. \tag{19}$$

Putting Eq. (19) into Eq. (18) yields:

$$y^\mu = \frac{1}{\mu} \left( A^\top x^\mu - \frac{\mathbf{1}_n^\top A^\top x^\mu - \mu}{n} \mathbf{1}_n \right) = \frac{1}{\mu} \left( A^\top \frac{\mathbf{1}_m}{m} - \frac{\mathbf{1}_n^\top A^\top \frac{\mathbf{1}_m}{m} - \mu}{n} \mathbf{1}_n \right),$$

where the second equality follows from $x^\mu = \frac{\mathbf{1}_m}{m}$. Multiplying this by $\frac{\mathbf{1}_m^\top}{m} A$, we have:

$$v^* = \frac{1}{\mu}\left(\frac{1}{m^2}\left\|A^\top \mathbf{1}_m\right\|^2 - \frac{1}{m^2 n}(\mathbf{1}_n^\top A^\top \mathbf{1}_m)^2 + \mu\frac{1}{mn}\mathbf{1}_m^\top A\mathbf{1}_n\right), \tag{20}$$

where we used the assumption that $y^\mu = y^*$ and Eq. (13). Here, we have:

$$\frac{1}{m^2}\left\|A^\top \mathbf{1}_m\right\|^2 - \frac{1}{m^2 n}(\mathbf{1}_n^\top A^\top \mathbf{1}_m)^2 + \mu\frac{1}{mn}\mathbf{1}_m^\top A\mathbf{1}_n$$

$$= \left\|v^*\mathbf{1}_n + A^\top \frac{\mathbf{1}_m}{m} - v^*\mathbf{1}_n\right\|^2 - \frac{1}{n}\left(\mathbf{1}_n^\top\left(v^*\mathbf{1}_n + A^\top \frac{\mathbf{1}_m}{m} - v^*\mathbf{1}_n\right)\right)^2 + \frac{\mu}{n}\mathbf{1}_n^\top\left(v^*\mathbf{1}_n + A^\top \frac{\mathbf{1}_m}{m} - v^*\mathbf{1}_n\right)$$

$$= n(v^*)^2 + \left\|A^\top \frac{\mathbf{1}_m}{m} - v^*\mathbf{1}_n\right\|^2 + 2v^*\mathbf{1}_n^\top\left(A^\top \frac{\mathbf{1}_m}{m} - v^*\mathbf{1}_n\right)$$

$$\quad - n(v^*)^2 - \frac{1}{n}\left(\mathbf{1}_n^\top\left(A^\top \frac{\mathbf{1}_m}{m} - v^*\mathbf{1}_n\right)\right)^2 - 2v^*\mathbf{1}_n^\top\left(A^\top \frac{\mathbf{1}_m}{m} - v^*\mathbf{1}_n\right) + \mu v^* + \frac{\mu}{n}\mathbf{1}_n^\top\left(A^\top \frac{\mathbf{1}_m}{m} - v^*\mathbf{1}_n\right)$$

$$= \mu v^* + \left\|A^\top \frac{\mathbf{1}_m}{m} - v^*\mathbf{1}_n\right\|^2 - \frac{1}{n}\left(\mathbf{1}_n^\top\left(A^\top \frac{\mathbf{1}_m}{m} - v^*\mathbf{1}_n\right)\right)^2 + \frac{\mu}{n}\mathbf{1}_n^\top\left(A^\top \frac{\mathbf{1}_m}{m} - v^*\mathbf{1}_n\right)$$

$$= \mu v^* + \left(A^\top \frac{\mathbf{1}_m}{m} - v^*\mathbf{1}_n\right)^\top\left(\mathbb{I} - \frac{1}{n}\mathbf{1}_n\mathbf{1}_n^\top\right)\left(A^\top \frac{\mathbf{1}_m}{m} - v^*\mathbf{1}_n\right) + \frac{\mu}{n}\mathbf{1}_n^\top\left(A^\top \frac{\mathbf{1}_m}{m} - v^*\mathbf{1}_n\right). \tag{21}$$

Here, since $A^\top \frac{\mathbf{1}_m}{m} = \frac{\mathbf{1}_n^\top A^\top x^\mu - \mu}{n}\mathbf{1}_n + \mu y^\mu = \frac{\frac{1}{m}\mathbf{1}_m^\top A\mathbf{1}_n - \mu}{n}\mathbf{1}_n + \mu y^*$ from Eq. (18) and Eq. (19), we get:

$$\left(A^\top \frac{\mathbf{1}_m}{m} - v^*\mathbf{1}_n\right)^\top\left(\mathbb{I} - \frac{1}{n}\mathbf{1}_n\mathbf{1}_n^\top\right)\left(A^\top \frac{\mathbf{1}_m}{m} - v^*\mathbf{1}_n\right) + \frac{\mu}{n}\mathbf{1}_n^\top\left(A^\top \frac{\mathbf{1}_m}{m} - v^*\mathbf{1}_n\right)$$

$$= \left(\left(\frac{\frac{1}{m}\mathbf{1}_m^\top A\mathbf{1}_n - \mu}{n} - v^*\right)\mathbf{1}_n + \mu y^*\right)^\top\left(\mathbb{I} - \frac{1}{n}\mathbf{1}_n\mathbf{1}_n^\top\right)\left(\left(\frac{\frac{1}{m}\mathbf{1}_m^\top A\mathbf{1}_n - \mu}{n} - v^*\right)\mathbf{1}_n + \mu y^*\right)$$

$$\quad + \frac{\mu}{n}\mathbf{1}_n^\top\left(\left(\frac{\frac{1}{m}\mathbf{1}_m^\top A\mathbf{1}_n - \mu}{n} - v^*\right)\mathbf{1}_n + \mu y^*\right)$$

$$= \mu\left(\frac{\frac{1}{m}\mathbf{1}_m^\top A\mathbf{1}_n - \mu}{n} - v^*\right) + \frac{\mu^2}{n} + \mu\left(\frac{\frac{1}{m}\mathbf{1}_m^\top A\mathbf{1}_n - \mu}{n} - v^*\right)\mathbf{1}_n^\top\left(\mathbb{I} - \frac{1}{n}\mathbf{1}_n\mathbf{1}_n^\top\right)y^*$$

$$\quad + \mu\left(\frac{\frac{1}{m}\mathbf{1}_m^\top A\mathbf{1}_n - \mu}{n} - v^*\right)(y^*)^\top\left(\mathbb{I} - \frac{1}{n}\mathbf{1}_n\mathbf{1}_n^\top\right)\mathbf{1}_n + \mu^2(y^*)^\top\left(\mathbb{I} - \frac{1}{n}\mathbf{1}_n\mathbf{1}_n^\top\right)y^*$$

$$= \mu\left(\frac{\frac{1}{m}\mathbf{1}_m^\top A\mathbf{1}_n - \mu}{n} - v^*\right) + \mu^2\left\|y^*\right\|^2$$

$$= \mu\left(\mu\left(\left\|y^*\right\|^2 - \frac{1}{n}\right) + \left(\frac{\mathbf{1}_m}{m}\right)^\top A\left(\frac{\mathbf{1}_n}{n}\right) - v^*\right). \tag{22}$$

By combining Eq. (20), Eq. (21), and Eq. (22), we have:

$$\mu\left(\left\|y^*\right\|^2 - \frac{1}{n}\right) + \left(\frac{\mathbf{1}_m}{m}\right)^\top A\left(\frac{\mathbf{1}_n}{n}\right) - v^* = 0.$$

Therefore, if $y^\mu = y^*$, then $\mu$ must satisfy:

$$\mu = \frac{v^* - \left(\frac{\mathbf{1}_m}{m}\right)^\top A\left(\frac{\mathbf{1}_n}{n}\right)}{\left(\left\|y^*\right\|^2 - \frac{1}{n}\right)},$$

and this is equivalent to:

$$\mu \neq \frac{v^* - \left(\frac{\mathbf{1}_m}{m}\right)^\top A \left(\frac{\mathbf{1}_n}{n}\right)}{\left(\|y^*\|^2 - \frac{1}{n}\right)} \Rightarrow y^\mu \neq y^*.$$

By a similar argument, in terms of player $x$, we can conclude that:

$$\mu \neq \frac{\left(\frac{\mathbf{1}_m}{m}\right)^\top A \left(\frac{\mathbf{1}_n}{n}\right) - v^*}{\left(\|x^*\|^2 - \frac{1}{m}\right)} \Rightarrow x^\mu \neq x^*.$$

$\square$

## F  PROOF OF THEOREM 3.1

*Proof of Theorem 3.1.* Let us define the function $g_{\mathrm{asym}}^\mu : \mathcal{X} \to \mathbb{R}$:

$$g_{\mathrm{asym}}^\mu(x) := \max_{y \in \mathcal{Y}} x^\top A y + \frac{\mu}{2} \|x\|^2.$$

Here, we introduce the following property of the function $\max_{y \in \mathcal{Y}} x^\top A y$:

**Lemma F.1** (Claim 1-5 in Theorem 5 of Wei et al. (2021))**.** *There exists a positive constant $\alpha > 0$ such that:*

$$\forall x \in \mathcal{X}, \max_{y \in \mathcal{Y}} x^\top A y - v^* \geq \alpha \|x - \Pi_{\mathcal{X}^*}(x)\|,$$

*where $\alpha$ depends only on $\mathcal{X}^*$.*

By using Lemma F.1, we have for any $x \in \mathcal{X}$:

$$\max_{y \in \mathcal{Y}} \left(\Pi_{\mathcal{X}^*}(x)\right)^\top A y - \max_{y \in \mathcal{Y}} x^\top A y = v^* - \max_{y \in \mathcal{Y}} x^\top A y \leq -\alpha \|x - \Pi_{\mathcal{X}^*}(x)\|.$$

On the other hand, for any $x^* \in \mathcal{X}^*$ and $x \in \mathcal{X} \setminus \mathcal{X}^*$, we have:

$$\begin{aligned}
\frac{\mu}{2} \|x^*\|^2 - \frac{\mu}{2} \|x\|^2 &= \frac{\mu}{2} \langle x^* - x, x^* + x \rangle \\
&= \frac{\mu}{2} \langle x^* - x, -x^* + x + 2x^* \rangle \\
&\leq \mu \langle x^* - x, x^* \rangle \\
&\leq \mu \|x^* - x\| \|x^*\|.
\end{aligned}$$

Summing up these inequalities, we obtain:

$$\begin{aligned}
g_{\mathrm{asym}}^\mu \left(\Pi_{\mathcal{X}^*}(x)\right) - g_{\mathrm{asym}}^\mu(x) &\leq -\alpha \|x - \Pi_{\mathcal{X}^*}(x)\| + \mu \|x - \Pi_{\mathcal{X}^*}(x)\| \|\Pi_{\mathcal{X}^*}(x)\| \\
&= -\left(\alpha - \mu \|\Pi_{\mathcal{X}^*}(x)\|\right) \|x - \Pi_{\mathcal{X}^*}(x)\| \\
&\leq -\left(\alpha - \mu \max_{x \in \mathcal{X}} \|x\|\right) \|x - \Pi_{\mathcal{X}^*}(x)\|.
\end{aligned}$$

Hence, under the assumption that $\mu < \frac{\alpha}{\max_{x \in \mathcal{X}} \|x\|}$, we have for any $x \in \mathcal{X} \setminus \mathcal{X}^*$:

$$g_{\mathrm{asym}}^\mu \left(\Pi_{\mathcal{X}^*}(x)\right) < g_{\mathrm{asym}}^\mu(x).$$

Thus, every $x \in \mathcal{X} \setminus \mathcal{X}^*$ is dominated by some equilibrium strategy $x^* \in \mathcal{X}^*$ with respect to the value of $g_{\mathrm{asym}}^\mu$. Therefore, we conclude that:

$$x^\mu \in \arg\min_{x \in \mathcal{X}} g_{\mathrm{asym}}^\mu(x) \Leftrightarrow x^\mu \in \arg\min_{x \in \mathcal{X}} \max_{y \in \mathcal{Y}} \left\{ x^\top A y + \frac{\mu}{2} \|x\|^2 \right\} \Leftrightarrow x^\mu \in \mathcal{X}^*.$$

$\square$

## G   PROOFS FOR THEOREM 4.1

### G.1   PROOF OF THEOREM 4.1

*Proof of Theorem 4.1.* First, we have for any vectors $a, b, c$:

$$\frac{1}{2} \left\| a - b \right\|^2 - \frac{1}{2} \left\| a - c \right\|^2 + \frac{1}{2} \left\| b - c \right\|^2 = \langle c - b, a - b \rangle. \tag{23}$$

From Eq. (23), we have for any $t \geq 1$:

$$\frac{1}{2} \left\| x^\mu - x^{t+1} \right\|^2 - \frac{1}{2} \left\| x^\mu - x^t \right\|^2 + \frac{1}{2} \left\| x^{t+1} - x^t \right\|^2 = \langle x^t - x^{t+1}, x^\mu - x^{t+1} \rangle \tag{24}$$

Here, we can rewrite the update rule as follows:

$$x^{t+1} = \arg\min_{p \in \mathcal{X}} \left\{ \eta \left\langle Ay^t + \mu x^t, p \right\rangle + \frac{1}{2} \left\| p - x^t \right\|^2 \right\},$$
$$y^{t+1} = \arg\min_{p \in \mathcal{Y}} \left\{ -\eta \left\langle A^\top x^{t+1}, p \right\rangle + \frac{1}{2} \left\| p - y^t \right\|^2 \right\}. \tag{25}$$

From the first-order optimality condition for $x^{t+1}$ in Eq. (25), we have for any $t \geq 1$:

$$\left\langle \eta Ay^t + \eta \mu x^t + x^{t+1} - x^t, x^{t+1} - x^\mu \right\rangle \leq 0. \tag{26}$$

Combining Eq. (24) and Eq. (26) yields:

$$\frac{1}{2} \left\| x^\mu - x^{t+1} \right\|^2 - \frac{1}{2} \left\| x^\mu - x^t \right\|^2 + \frac{1}{2} \left\| x^{t+1} - x^t \right\|^2$$
$$\leq \eta \left\langle Ay^t + \mu x^t, x^\mu - x^{t+1} \right\rangle$$
$$= \eta \left\langle Ay^{t+1} + \mu x^{t+1}, x^\mu - x^{t+1} \right\rangle + \eta \left\langle Ay^t - Ay^{t+1} + \mu(x^t - x^{t+1}), x^\mu - x^{t+1} \right\rangle$$
$$= \eta \left\langle Ay^\mu + \mu x^\mu, x^\mu - x^{t+1} \right\rangle + \eta \left\langle Ay^t - Ay^{t+1} + \mu(x^t - x^{t+1}), x^\mu - x^{t+1} \right\rangle$$
$$+ \eta \left\langle Ay^{t+1} - Ay^\mu, x^\mu - x^{t+1} \right\rangle - \eta \mu \left\| x^\mu - x^{t+1} \right\|^2. \tag{27}$$

On the other hand, from the first-order optimality condition for $x^\mu$ in Eq. (4), we get:

$$\left\langle Ay^\mu + \mu x^\mu, x^\mu - x^{t+1} \right\rangle \leq 0. \tag{28}$$

By combining Eq. (27) and Eq. (28), we have for any $t \geq 1$:

$$\frac{1}{2} \left\| x^\mu - x^{t+1} \right\|^2 - \frac{1}{2} \left\| x^\mu - x^t \right\|^2 + \frac{1}{2} \left\| x^{t+1} - x^t \right\|^2$$
$$\leq -\eta \mu \left\| x^\mu - x^{t+1} \right\|^2 + \eta \left\langle Ay^t - Ay^{t+1}, x^\mu - x^{t+1} \right\rangle + \eta \mu \left\langle x^t - x^{t+1}, x^\mu - x^{t+1} \right\rangle$$
$$+ \eta \left\langle Ay^{t+1} - Ay^\mu, x^\mu - x^{t+1} \right\rangle. \tag{29}$$

Similar to Eq. (24), we have for any $t \geq 1$:

$$\frac{1}{2} \left\| y^\mu - y^{t+1} \right\|^2 - \frac{1}{2} \left\| y^\mu - y^t \right\|^2 + \frac{1}{2} \left\| y^{t+1} - y^t \right\|^2 = \left\langle y^t - y^{t+1}, y^\mu - y^{t+1} \right\rangle, \tag{30}$$

and from the first-order optimality condition for $y^{t+1}$ in Eq. (25), we have for any $t \geq 1$:

$$\left\langle -\eta A^\top x^{t+1} + y^{t+1} - y^t, y^{t+1} - y^\mu \right\rangle \leq 0. \tag{31}$$

By combining Eq. (30) and Eq. (31), we get:

$$\frac{1}{2} \left\| y^\mu - y^{t+1} \right\|^2 - \frac{1}{2} \left\| y^\mu - y^t \right\|^2 + \frac{1}{2} \left\| y^{t+1} - y^t \right\|^2$$
$$\leq -\eta \left\langle A^\top x^{t+1}, y^\mu - y^{t+1} \right\rangle$$
$$= -\eta \left\langle A^\top x^\mu, y^\mu - y^{t+1} \right\rangle - \eta \left\langle A^\top x^{t+1} - A^\top x^\mu, y^\mu - y^{t+1} \right\rangle$$
$$\leq -\eta \left\langle A^\top x^{t+1} - A^\top x^\mu, y^\mu - y^{t+1} \right\rangle, \tag{32}$$

where the last inequality stems from Eq. (5).

Summing up Eq. (29) and Eq. (32), we have for any $t \geq 1$:

$$\frac{1}{2} \left\| z^\mu - z^{t+1} \right\|^2 - \frac{1}{2} \left\| z^\mu - z^t \right\|^2 + \frac{1}{2} \left\| z^{t+1} - z^t \right\|^2$$

$$\leq -\eta\mu \left\| x^\mu - x^{t+1} \right\|^2 + \eta\mu \left\langle x^t - x^{t+1}, x^\mu - x^{t+1} \right\rangle + \eta \left\langle Ay^t - Ay^{t+1}, x^\mu - x^{t+1} \right\rangle$$

$$\leq -\eta\mu \left\| x^\mu - x^{t+1} \right\|^2 + \frac{1}{4} \left\| x^t - x^{t+1} \right\|^2 + \eta^2\mu^2 \left\| x^\mu - x^{t+1} \right\|^2 + \frac{1}{4} \left\| y^t - y^{t+1} \right\|^2 + \eta^2 \|A\|^2 \left\| x^\mu - x^{t+1} \right\|^2$$

$$= -\left( \eta\mu - \eta^2(\mu^2 + \|A\|^2) \right) \left\| x^\mu - x^{t+1} \right\|^2 + \frac{1}{4} \left\| z^t - z^{t+1} \right\|^2 .$$

Hence, under the assumption that $\eta < \frac{\mu}{2(\mu^2 + \|A\|^2)}$, we have for any $y^\mu \in \mathcal{Y}^\mu$:

$$\frac{1}{2} \left\| z^\mu - z^{t+1} \right\|^2 - \frac{1}{2} \left\| z^\mu - z^t \right\|^2 \leq -\frac{\eta\mu}{2} \left\| x^\mu - x^{t+1} \right\|^2 - \frac{1}{4} \left\| z^t - z^{t+1} \right\|^2 . \tag{33}$$

In terms of the path length $\left\| z^t - z^{t+1} \right\|$, we can we derive the following lower bound:

**Lemma G.1.** *For any $t \geq 1$, we have:*

$$\left\| z^t - z^{t+1} \right\| \geq \frac{\eta}{D(1 + \eta\|A\|)} \left( \frac{\mu}{2} \left\| x^\mu - x^{t+1} \right\|^2 + \frac{\beta}{2\mu} \left\| \Pi_{\mathcal{Y}^\mu}(y^{t+1}) - y^{t+1} \right\|^2 \right),$$

*where $\beta > 0$ is a positive constant depending only on $\mathcal{Y}^\mu$.*

From Eq. (33) and Lemma G.1, we obtain for any $y^\mu \in \mathcal{Y}^\mu$:

$$\frac{1}{2} \left\| z^\mu - z^{t+1} \right\|^2 - \frac{1}{2} \left\| z^\mu - z^t \right\|^2$$

$$\leq -\frac{\eta\mu}{2} \left\| x^\mu - x^{t+1} \right\|^2 - \frac{1}{4} \left\| z^t - z^{t+1} \right\|^2$$

$$\leq -\frac{\eta\mu}{2} \left\| x^\mu - x^{t+1} \right\|^2 - \frac{\eta^2}{4D^2(1 + \eta\|A\|)^2} \left( \frac{\mu^2}{4} \left\| x^\mu - x^{t+1} \right\|^4 + \frac{\beta^2}{4\mu^2} \left\| \Pi_{\mathcal{Y}^\mu}(y^{t+1}) - y^{t+1} \right\|^4 \right)$$

$$\leq -\frac{\eta^2 \min\left( \mu^2, \frac{\beta^2}{\mu^2} \right)}{16D^2(1 + \eta\|A\|)^2} \left( \left\| x^\mu - x^{t+1} \right\|^4 + \left\| \Pi_{\mathcal{Y}^\mu}(y^{t+1}) - y^{t+1} \right\|^4 \right)$$

$$\leq -\frac{\eta^2 \min\left( \mu^2, \frac{\beta^2}{\mu^2} \right)}{32D^2(1 + \eta\|A\|)^2} \left( \left\| x^\mu - x^{t+1} \right\|^2 + \left\| \Pi_{\mathcal{Y}^\mu}(y^{t+1}) - y^{t+1} \right\|^2 \right)^2 .$$

where the second inequality follows from the fact that $(a + b)^2 \geq a^2 + b^2$ for any $a, b \geq 0$, and the fourth inequality follows from the fact that $a^2 + b^2 \geq \frac{1}{2}(a + b)^2$ for any $a, b \geq 0$. Then, under the assumption that $\eta < \frac{\mu}{2(\mu^2 + \|A\|^2)}$, we have for any $y^\mu \in \mathcal{Y}^\mu$:

$$\frac{1}{2} \left\| z^\mu - z^{t+1} \right\|^2 - \frac{1}{2} \left\| z^\mu - z^t \right\|^2$$

$$\leq -\frac{2\eta^2(\mu^2 + \|A\|^2) \min\left( \mu^2, \frac{\beta^2}{\mu^2} \right)}{32D^2(2(\mu^2 + \|A\|^2) + \mu\|A\|)^2} \left( \left\| x^\mu - x^{t+1} \right\|^2 + \left\| \Pi_{\mathcal{Y}^\mu}(y^{t+1}) - y^{t+1} \right\|^2 \right)^2$$

$$\leq -\frac{\eta^2 (\mu + \|A\|)^2 \min\left( \mu^2, \frac{\beta^2}{\mu^2} \right)}{128D^2 (\mu + \|A\|)^4} \left( \left\| x^\mu - x^{t+1} \right\|^2 + \left\| \Pi_{\mathcal{Y}^\mu}(y^{t+1}) - y^{t+1} \right\|^2 \right)^2$$

$$= -\frac{\eta^2 \min\left( \mu^2, \frac{\beta^2}{\mu^2} \right)}{128D^2 (\mu + \|A\|)^2} \left( \left\| x^\mu - x^{t+1} \right\|^2 + \left\| \Pi_{\mathcal{Y}^\mu}(y^{t+1}) - y^{t+1} \right\|^2 \right)^2 .$$

By choosing $y^\mu = \Pi_{\mathcal{Y}^\mu}(y^t)$, we get for any $t \geq 1$:

$$\left\|x^\mu - x^{t+1}\right\|^2 + \left\|\Pi_{\mathcal{Y}^\mu}(y^{t+1}) - y^{t+1}\right\|^2$$

$$\leq \left\|x^\mu - x^{t+1}\right\|^2 + \left\|\Pi_{\mathcal{Y}^\mu}(y^t) - y^{t+1}\right\|^2$$

$$\leq \left\|x^\mu - x^t\right\|^2 + \left\|\Pi_{\mathcal{Y}^\mu}(y^t) - y^t\right\|^2 - \frac{\eta^2 \min\left(\mu^2, \frac{\beta^2}{\mu^2}\right)}{128D^2\left(\mu + \|A\|\right)^2}\left(\left\|x^\mu - x^{t+1}\right\|^2 + \left\|\Pi_{\mathcal{Y}^\mu}(y^{t+1}) - y^{t+1}\right\|^2\right)^2.$$

To obtain the convergence rate of $\left\|x^\mu - x^t\right\|^2 + \left\|\Pi_{\mathcal{Y}^\mu}(y^t) - y^t\right\|^2$, we introduce the following lemma from Wei et al. (2021):

**Lemma G.2** (Lemma 12 of Wei et al. (2021)). *Consider a non-negative sequence $\{B_t\}_{t\geq 1}$ that satisfies for some $q > 0$,*

$$B_{t+1} \leq B_t - qB_{t+1}^2, \ t \geq 1$$

*and*

$$2qB_1 \leq 1.$$

*Then, $B_t \leq \frac{\max\left(B_1, \frac{2}{q}\right)}{t}$.*

Under the assumption that $\eta \leq \frac{8(\mu + \|A\|)}{D\min\left(\mu, \frac{\beta}{\mu}\right)}$:

$$2\frac{\eta^2 \min\left(\mu^2, \frac{\beta^2}{\mu^2}\right)}{128D^2\left(\mu + \|A\|\right)^2}\left(\left\|x^\mu - x^1\right\|^2 + \left\|\Pi_{\mathcal{Y}^\mu}(y^1) - y^1\right\|^2\right)^2 \leq \frac{\eta^2 D^2 \min\left(\mu^2, \frac{\beta^2}{\mu^2}\right)}{64\left(\mu + \|A\|\right)^2} \leq 1.$$

Therefore, we can apply Lemma G.2 with $q = \frac{\eta^2 \min\left(\mu^2, \frac{\beta^2}{\mu^2}\right)}{128D^2(\mu + \|A\|)^2}$, and then we have:

$$\left\|x^\mu - x^t\right\|^2 + \left\|\Pi_{\mathcal{Y}^\mu}(y^t) - y^t\right\|^2 \leq \frac{\max\left(\left\|x^\mu - x^1\right\|^2 + \left\|\Pi_{\mathcal{Y}^\mu}(y^1) - y^1\right\|^2, \frac{256D^2(\mu + \|A\|)^2}{\eta^2 \min\left(\mu^2, \frac{\beta^2}{\mu^2}\right)}\right)}{t}$$

$$\leq \frac{D^2 \max\left(1, \frac{256(\mu + \|A\|)^2}{\eta^2 \min\left(\mu^2, \frac{\beta^2}{\mu^2}\right)}\right)}{t}$$

$$\leq \frac{256D^2\left(\mu + \|A\|\right)^2}{\eta^2 t \min\left(\mu^2, \frac{\beta^2}{\mu^2}\right)}.$$

$\square$

## G.2 Proof of Lemma G.1

*Proof of Lemma G.1.* We first derive the lower bound on the path length $\left\|z^t - z^{t+1}\right\|$ by the suboptimality gap:

**Lemma G.3.** *For any $x \in \mathcal{X}$, $y^\mu \in \mathcal{Y}^\mu$, and $t \geq 1$, we have:*

$$\frac{\eta}{1 + \eta\|A\|} \frac{\langle Ay^{t+1} + \mu x^{t+1}, x^{t+1} - x\rangle - \langle A^\top x^{t+1}, y^{t+1} - y^\mu\rangle}{\|x^{t+1} - x\| + \|y^{t+1} - y^\mu\|} \leq \left\|z^{t+1} - z^t\right\|.$$

We also derive the lower bound on the suboptimality gap for any strategy profile $(x, y) \in \mathcal{X} \times \mathcal{Y}$:

**Lemma G.4.** *For any $x \in \mathcal{X}$, $y \in \mathcal{Y}$, and $y^\mu \in \mathcal{Y}^\mu$, we have:*

$$\max_{x' \in \mathcal{X}} \left(\langle Ay + \mu x, x - x'\rangle - \langle A^\top x, y - y^\mu\rangle\right) \geq \frac{\mu}{2}\|x - x^\mu\|^2 + \frac{1}{2\mu}\|A(y^\mu - y)\|^2.$$

By combining Lemmas G.3, and G.4, we have:

$$\left\| z^{t+1} - z^t \right\|$$

$$\geq \frac{\eta}{1 + \eta \|A\|} \max_{x \in \mathcal{X}} \frac{\langle Ay^{t+1} + \mu x^{t+1}, x^{t+1} - x \rangle - \langle A^\top x^{t+1}, y^{t+1} - y^\mu \rangle}{\|x^{t+1} - x\| + \|y^{t+1} - y^\mu\|}$$

$$\geq \frac{\eta}{D(1 + \eta \|A\|)} \max_{x \in \mathcal{X}} \left( \langle Ay^{t+1} + \mu x^{t+1}, x^{t+1} - x \rangle - \langle A^\top x^{t+1}, y^{t+1} - y^\mu \rangle \right)$$

$$\geq \frac{\eta}{D(1 + \eta \|A\|)} \left( \frac{\mu}{2} \left\| x^{t+1} - x^\mu \right\|^2 + \frac{1}{2\mu} \left\| A(y^\mu - y^{t+1}) \right\|^2 \right).$$

Here, for any maximin strategy $y^\mu \in \mathcal{Y}^\mu$ in the perturbed game, $Ay^\mu$ can be written as $Ay^\mu = b^*$ by using some vector $b^*$:

**Lemma G.5.** *There exists a vector $b^*$ such that $b^* = Ay^\mu$ for all $y^\mu \in \mathcal{Y}^\mu$. Furthermore, for any strategy $\tilde{y} \in \mathcal{Y}$, we have:*

$$\tilde{y} \in \mathcal{Y}^\mu \Leftrightarrow A\tilde{y} = b^*.$$

From Lemma G.5, we can rewrite $\left\| A(y^\mu - y^{t+1}) \right\|^2 = \left\| Ay^{t+1} - b^* \right\|^2$. The following lemma demonstrates that this term can be lower bounded by the distance between $y^{t+1}$ and the set of maximin strategies $\mathcal{Y}^\mu$:

**Lemma G.6.** *There exists a positive constant $\beta > 0$ such that for any $y \in \mathcal{Y}$:*

$$\left\| Ay - b^* \right\|^2 \geq \beta \left\| y - \Pi_{\mathcal{Y}^\mu}(y) \right\|^2.$$

Therefore, we conclude that:

$$\left\| z^{t+1} - z^t \right\| \geq \frac{\eta}{D(1 + \eta \|A\|)} \left( \frac{\mu}{2} \left\| x^{t+1} - x^\mu \right\|^2 + \frac{\beta}{2\mu} \left\| \Pi_{\mathcal{Y}^\mu}(y^{t+1}) - y^{t+1} \right\|^2 \right).$$

$$\square$$

## G.3 PROOF OF LEMMA G.3

*Proof of Lemma G.3.* From the first-order optimality condition for $x^{t+1}$, we have for any $x \in \mathcal{X}$ and $t \geq 1$:

$$\langle \eta Ay^t + \eta \mu x^t + x^{t+1} - x^t, x - x^{t+1} \rangle \geq 0.$$

Rearranging the terms yields:

$$\langle x^{t+1} - x^t, x - x^{t+1} \rangle$$

$$\geq \eta \langle Ay^t + \mu x^t, x^{t+1} - x \rangle$$

$$= \eta \langle Ay^{t+1} + \mu x^{t+1}, x^{t+1} - x \rangle + \eta \langle Ay^t - Ay^{t+1}, x^{t+1} - x \rangle + \eta \mu \langle x^t - x^{t+1}, x^{t+1} - x \rangle. \quad (34)$$

Similarly, from the first-order optimality condition for $y^{t+1}$, we have for any $t \geq 1$:

$$\langle -\eta A^\top x^{t+1} + y^{t+1} - y^t, y^\mu - y^{t+1} \rangle \geq 0.$$

Rearranging the terms yields:

$$\langle y^{t+1} - y^t, y^\mu - y^{t+1} \rangle \geq -\eta \langle A^\top x^{t+1}, y^{t+1} - y^\mu \rangle. \quad (35)$$

By combining Eq. (34) and Eq. (35) and Cauchy-Schwarz inequality, we have for any $t \geq 1$:

$$\eta \langle Ay^{t+1} + \mu x^{t+1}, x^{t+1} - x \rangle - \eta \langle A^\top x^{t+1}, y^{t+1} - y^\mu \rangle$$

$$\leq \eta \langle Ay^t - Ay^{t+1}, x - x^{t+1} \rangle + (1 - \eta\mu) \langle x^t - x^{t+1}, x^{t+1} - x \rangle + \langle y^{t+1} - y^t, y^\mu - y^{t+1} \rangle$$

$$\leq \eta \|A\| \left\| y^t - y^{t+1} \right\| \left\| x - x^{t+1} \right\| + (1 - \eta\mu) \left\| x^t - x^{t+1} \right\| \left\| x^{t+1} - x \right\| + \left\| y^{t+1} - y^t \right\| \left\| y^\mu - y^{t+1} \right\|$$

$$\leq \left( \eta \|A\| \left\| y^t - y^{t+1} \right\| + (1 - \eta\mu) \left\| x^t - x^{t+1} \right\| \right) \left\| x - x^{t+1} \right\| + \left\| y^{t+1} - y^t \right\| \left\| y^\mu - y^{t+1} \right\|.$$

Therefore, we have for any $t \geq 1$:

$$\eta \frac{\langle Ay^{t+1} + \mu x^{t+1}, x^{t+1} - x \rangle - \langle A^\top x^{t+1}, y^{t+1} - y^\mu \rangle}{\|x^{t+1} - x\| + \|y^{t+1} - y^\mu\|}$$

$$\leq \frac{\left(\eta\|A\| \|y^t - y^{t+1}\| + (1 - \eta\mu) \|x^t - x^{t+1}\|\right) \|x - x^{t+1}\|}{\|x^{t+1} - x\| + \|y^{t+1} - y^\mu\|} + \frac{\|y^{t+1} - y^t\| \|y^\mu - y^{t+1}\|}{\|x^{t+1} - x\| + \|y^{t+1} - y^\mu\|}$$

$$\leq \eta\|A\| \|y^t - y^{t+1}\| + (1 - \eta\mu) \|x^t - x^{t+1}\| + \|y^{t+1} - y^t\|$$

$$\leq (1 + \eta\|A\|) \|z^t - z^{t+1}\|.$$

$\square$

### G.4 Proof of Lemma G.4

*Proof of Lemma G.4.* From the strongly convexity of the function $\frac{1}{2}\|x\|^2$, we have for any $(x, y) \in \mathcal{X} \times \mathcal{Y}$ and $x' \in \mathcal{X}$:

$$x^\top Ay + \frac{\mu}{2} \|x\|^2 - (x')^\top Ay - \frac{\mu}{2} \|x'\|^2 \leq \langle Ay + \mu x, x - x' \rangle - \frac{\mu}{2} \|x - x'\|^2.$$

Moreover, from the linearity of the function $x^\top Ay$ with respect to $y \in \mathcal{Y}$, we get for any $(x, y) \in \mathcal{X} \times \mathcal{Y}$ and $y^\mu \in \mathcal{Y}^\mu$:

$$x^\top Ay^\mu + \frac{\mu}{2} \|x\|^2 - x^\top Ay - \frac{\mu}{2} \|x\|^2 = \langle A^\top x, y^\mu - y \rangle.$$

Summing up the above two inequalities, we get for any $y^\mu \in \mathcal{Y}^\mu$:

$$\langle Ay + \mu x, x - x' \rangle - \langle A^\top x, y - y^\mu \rangle$$

$$\geq x^\top Ay^\mu + \frac{\mu}{2} \|x\|^2 - (x')^\top Ay - \frac{\mu}{2} \|x'\|^2 + \frac{\mu}{2} \|x - x'\|^2$$

$$\geq x^\top Ay^\mu + \frac{\mu}{2} \|x\|^2 - \left((x^\mu)^\top Ay^\mu + \frac{\mu}{2} \|x^\mu\|^2\right)$$

$$+ \left((x^\mu)^\top Ay^\mu + \frac{\mu}{2} \|x^\mu\|^2\right) - \left((x')^\top Ay + \frac{\mu}{2} \|x'\|^2\right)$$

$$\geq \frac{\mu}{2} \|x - x^\mu\|^2 + \left((x^\mu)^\top Ay^\mu + \frac{\mu}{2} \|x^\mu\|^2\right) - \left((x')^\top Ay + \frac{\mu}{2} \|x'\|^2\right).$$

Hence, we obtain:

$$\max_{x' \in \mathcal{X}} \langle Ay + \mu x, x - x' \rangle - \langle A^\top x, y - y^\mu \rangle$$

$$\geq \frac{\mu}{2} \|x - x^\mu\|^2 + \left((x^\mu)^\top Ay^\mu) + \frac{\mu}{2} \|x^\mu\|^2\right) - \min_{x' \in \mathcal{X}} \left((x')^\top Ay + \frac{\mu}{2} \|x'\|^2\right)$$

$$= \frac{\mu}{2} \|x - x^\mu\|^2 + \max_{\tilde{y} \in \mathcal{Y}} \min_{\tilde{x} \in \mathcal{X}} \left((\tilde{x})^\top A\tilde{y} + \frac{\mu}{2} \|\tilde{x}\|^2\right) - \min_{x' \in \mathcal{X}} \left((x')^\top Ay + \frac{\mu}{2} \|x'\|^2\right).$$

Here, the suboptimality gap of the strategy for player $x$ can be lower bounded as follows:

**Lemma G.7.** *For any $y \in \mathcal{Y}$ and $y^\mu \in \mathcal{Y}^\mu$, we have:*

$$\max_{y' \in \mathcal{Y}} \min_{x' \in \mathcal{X}} \left((x')^\top Ay' + \frac{\mu}{2} \|x'\|^2\right) - \min_{x' \in \mathcal{X}} \left((x')^\top Ay + \frac{\mu}{2} \|x'\|^2\right) \geq \frac{1}{2\mu} \|A(y^\mu - y)\|^2.$$

Combining this inequality with Lemma G.7, we have:

$$\max_{x' \in \mathcal{X}} \langle Ay + \mu x, x - x' \rangle - \langle A^\top x, y - y^\mu \rangle \geq \frac{\mu}{2} \|x - x^\mu\|^2 + \frac{1}{2\mu} \|A(y^\mu - y)\|^2.$$

$\square$

## G.5   PROOF OF LEMMA G.5

*Proof of Lemma G.5.* Let us define $\mathcal{B} := \{-Ay \mid y \in \mathcal{Y}\}$. We also define $h(x) := \frac{\mu}{2}\|x\|^2$ and its convex conjugate $h^*(p) := \max_{x \in \mathcal{X}}\{x^\top p - h(x)\}$. Since $h$ is a $\mu$-smooth function, the convex conjugate $h^*$ is $\frac{1}{\mu}$-strongly convex (Shalev-Shwartz, 2011, p. 149). Thus, $\arg\min_{b \in \mathcal{B}} h^*(b)$ is a singleton, and then we can write $b^* = \arg\min_{b \in \mathcal{B}} h^*(b)$. Hence, every $\tilde{y} \in \mathcal{Y}^\mu$ must satisfy $A\tilde{y} = b^*$. Thus, we have $\tilde{y} \in \mathcal{Y}^\mu \Rightarrow A\tilde{y} = b^*$.

Next, we show that $A\tilde{y} = b^* \Rightarrow \tilde{y} \in \mathcal{Y}^\mu$. We have for any $\tilde{y} \in \mathcal{Y}$ satisfying $A\tilde{y} = b^*$:

$$\forall y \in \mathcal{Y},\ (x^\mu)^\top A\tilde{y} = (x^\mu)^\top b^* \geq (x^\mu)^\top Ay,$$

since $(x^\mu)^\top b^* = (x^\mu)^\top Ay^\mu \geq (x^\mu)^\top Ay$. Thus, $\tilde{y}$ is the best response against $x^\mu$ in the perturbed game. Moreover, since $Ay^\mu = b^*$ for any $y^\mu \in \mathcal{Y}^\mu$ and $x^\mu$ is an equilibrium in the perturbed game, we have:

$$\forall x \in \mathcal{Y},\ (x^\mu)^\top b^* + \frac{\mu}{2}\|x^\mu\|^2 \leq x^\top b^* + \frac{\mu}{2}\|x\|^2.$$

Hence, since $b^* = A\tilde{y}$, we have:

$$\forall x \in \mathcal{X},\ (x^\mu)^\top A\tilde{y} + \frac{\mu}{2}\|x^\mu\|^2 \leq x^\top A\tilde{y} + \frac{\mu}{2}\|x\|^2,$$

and then $x^\mu$ is also a best response against $\tilde{y}$. Consequently, it holds that $(x^\mu, \tilde{y})$ is an equilibrium in the perturbed game, and $\tilde{y} \in \mathcal{Y}^\mu$. □

## G.6   PROOF OF LEMMA G.6

*Proof of Lemma G.6.* Since the statement clearly holds for $y \in \mathcal{Y}^\mu$, we consider an arbitrary strategy $y \in \mathcal{Y} \setminus \mathcal{Y}^\mu$. First, we have for any convex set $\mathcal{S}$:

$$s' - s \in \mathcal{N}_\mathcal{S}(s) \Leftrightarrow s = \Pi_\mathcal{S}(s').$$

Thus, defining $y^\mu = \Pi_{\mathcal{Y}^\mu}(y)$, we get:

$$\mathcal{N}_{\mathcal{Y}^\mu}(y^\mu) = \{\tilde{y} - \tilde{y}^\mu \mid \tilde{y} \in \mathbb{R}^n,\ \tilde{y}^\mu = \Pi_{\mathcal{Y}^\mu}(\tilde{y})\}. \tag{36}$$

On the other hand, since $\mathcal{Y}$ is a polytope and also a polyhedron, we can write $\mathcal{Y} = \{y \in \mathbb{R}^n \mid \langle d_i, y\rangle \leq e_i \text{ for } i = 1, \cdots, L\}$ for some $\{d_i\}_{i=1}^L$ and $\{e_i\}_{i=1}^L$. Hence, from Lemma G.5, $\mathcal{Y}^\mu$ can be represented as:

$$\mathcal{Y}^\mu = \{y \in \mathbb{R}^n \mid \langle d_i, y\rangle \leq e_i \text{ for } i = 1, \cdots, L,\ \langle a_i, y\rangle = b_i^* \text{ for } i = 1, \cdots, m\},$$

where $A = (a_1, \cdots, a_m)^\top$. Thus, $\mathcal{Y}^\mu$ is also a polytope. Hence, from Theorem 6.46 in Rockafellar & Wets (2009), $\mathcal{N}_{\mathcal{Y}^\mu}(y^\mu)$ can be written as:

$$\mathcal{N}_{\mathcal{Y}^\mu}(y^\mu) = \Bigg\{\sum_{i=1}^L p_i d_i + \sum_{i=1}^m q_i a_i - \sum_{i=1}^m r_i a_i \mid p_i \geq 0 \text{ for } i \in I(y^\mu),\ p_i = 0 \text{ for } i \notin I(y^\mu),$$

$$q_i \geq 0 \text{ for } i \in I^*(y^\mu), q_i = 0 \text{ for } i \notin I^*(y^\mu),$$

$$r_i \geq 0 \text{ for } i \in I^*(y^\mu),\ r_i = 0 \text{ for } i \notin I^*(y^\mu)\Bigg\},$$

where $I(y^\mu) := \{i \in [1, L] \mid \langle d_i, y^\mu\rangle = e_i\}$ and $I^*(y^\mu) := \{i \in [1, m] \mid \langle a_i, y^\mu\rangle = b_i^*\}$. Since $y^\mu \in \mathcal{Y}^\mu$, it holds that $i \in I^*(y^\mu)$ for all $i \in [1, m]$. Furthermore, without loss of generality, we assume that $\langle d_i, y^\mu\rangle = e_i$ for $i = 1, \cdots, l$ and $\langle d_i, y^\mu\rangle < e_i$ for $i = l+1, \cdots, L$ with some $l \leq L$. Then, we obtain:

$$\mathcal{N}_{\mathcal{Y}^\mu}(y^\mu) = \Bigg\{\sum_{i=1}^l p_i d_i + \sum_{i=1}^m (q_i - r_i)a_i \mid p_i \geq 0,\ q_i \geq 0,\ r_i \geq 0\Bigg\}. \tag{37}$$

Combining Eq. (36) and Eq. (37):

$$\mathcal{N}_{\mathcal{Y}^\mu}(y^\mu) = \{\tilde{y} - \tilde{y}^\mu \mid \tilde{y} \in \mathbb{R}^n, \ \tilde{y}^\mu = \Pi_{\mathcal{Y}^\mu}(\tilde{y})\}$$

$$= \left\{ \sum_{i=1}^{l} p_i d_i + \sum_{i=1}^{m} (q_i - r_i) a_i \mid p_i \geq 0, \ q_i \geq 0, \ r_i \geq 0 \right\}. \tag{38}$$

Since $y^\mu \in \Pi_{\mathcal{Y}^\mu}(y)$, we have $y - y^\mu \in \mathcal{N}_{\mathcal{Y}^\mu}(y^\mu)$. Thus, we can write $y - y^\mu = \sum_{i=1}^{l} p_i d_i + \sum_{i=1}^{m} (q_i - r_i) a_i$ with $p_i \geq 0$, $q_i \geq 0$, and $r_i \geq 0$. Moreover, we have for any $i = 1, \cdots, l$:

$$\langle d_i, y - y^\mu \rangle = \langle d_i, y \rangle - e_i \leq 0.$$

Hence, from Eq. (38), we obtain:

$$y - y^\mu \in \mathcal{M}(y^\mu) := \left\{ \sum_{i=1}^{l} p_i d_i + \sum_{i=1}^{m} (q_i - r_i) a_i \mid p_i \geq 0, \ q_i \geq 0, \ r_i \geq 0, \right.$$

$$\left. \left\langle d_i, \sum_{j=1}^{l} p_j d_j + \sum_{j=1}^{m} (q_j - r_j) a_j \right\rangle \leq 0 \text{ for } i \in [1, l] \right\}.$$

Since $\mathcal{M}(y^\mu)$ is a cone, we also have $\frac{y - y^\mu}{\|y - y^\mu\|} \in \mathcal{M}(y^\mu)$. Then, we have:

$$\frac{y - y^\mu}{\|y - y^\mu\|} \in \mathcal{P}(y^\mu) := \left\{ \sum_{i=1}^{l} p_i d_i + \sum_{i=1}^{m} (q_i - r_i) a_i \mid p_i \geq 0, \ q_i \geq 0, \ r_i \geq 0, \right.$$

$$\left\langle d_i, \sum_{j=1}^{l} p_j d_j + \sum_{j=1}^{m} (q_j - r_j) a_j \right\rangle \leq 0 \text{ for } i \in [1, l],$$

$$\left. \left\| \sum_{i=1}^{l} p_i d_i + \sum_{i=1}^{m} (q_i - r_i) a_i \right\|_\infty \leq 1 \right\}.$$

Since $\mathcal{P}(y^\mu)$ is a bounded polyhedron, it is also a polytope. Now, let us take an arbitrary vertex $v$ of $\mathcal{P}(y^\mu)$ and consider the following optimization problem:

$$\min_{p_i, q_i, r_i, C_v} C_v, \text{ s.t., } v = \sum_{i=1}^{l} p_i d_i + \sum_{i=1}^{m} (q_i - r_i) a_i, \ 0 \leq p_i \leq C_v, 0 \leq q_i \leq C_v, 0 \leq r_i \leq C_v.$$

Since $v \in \mathcal{M}(y^\mu)$, this optimization problem is always feasible and admits a finite solution $C_v < \infty$. Let us denote the optimal value of this optimization problem as $C_v^*$, and define $C_{y^\mu} = \max_{v \in \mathcal{V}(\mathcal{P}(y^\mu))} C_v^*$, where $\mathcal{V}(\mathcal{P}(y^\mu))$ is the set of all vertices of $\mathcal{P}(y^\mu)$. Since any $v \in \mathcal{P}(y^\mu)$ can be expressed as a convex combination of some points in $\mathcal{V}(\mathcal{P}(y^\mu))$, $v$ can be written as $\sum_{i=1}^{l} p_i a_i + \sum_{i=1}^{m} (q_i - r_i) a_i$ with $0 \leq p_i \leq C_{y^\mu}, 0 \leq q_i \leq C_{y^\mu}$, and $0 \leq r_i \leq C_{y^\mu}$. Thus, since $\frac{y - y^\mu}{\|y - y^\mu\|} \in \mathcal{P}(y^\mu)$, $\frac{y - y^\mu}{\|y - y^\mu\|}$ can be expressed as $\sum_{i=1}^{l} p_i d_i + \sum_{i=1}^{m} (q_i - r_i) a_i$ with $0 \leq p_i \leq C_{y^\mu}$, $0 \leq q_i \leq C_{y^\mu}$, and $0 \leq r_i \leq C_{y^\mu}$.

From the above discussion, we can write $y - y^\mu$ as $\sum_{i=1}^{l} p_i d_i + \sum_{i=1}^{m} (q_i - r_i) a_i$ with $0 \leq p_i \leq C_{y^\mu} \|y - y^\mu\|, 0 \leq q_i \leq C_{y^\mu} \|y - y^\mu\|$, and $0 \leq r_i \leq C_{y^\mu} \|y - y^\mu\|$. Then,

$$\sum_{i=1}^{l} p_i \langle d_i, y - y^\mu \rangle + \sum_{i=1}^{m} (q_i - r_i) \langle a_i, y - y^\mu \rangle$$

$$= \left\langle \sum_{i=1}^{l} p_i d_i + \sum_{i=1}^{m} (q_i - r_i) a_i, y - y^\mu \right\rangle = \|y - y^\mu\|^2. \tag{39}$$

Here, since $y - y^\mu \in \mathcal{M}(y^\mu)$ and $y - y^\mu = \sum_{i=1}^{l} p_i d_i + \sum_{i=1}^{m} (q_i - r_i) a_i$, we have:

$$\sum_{i=1}^{l} p_i \langle d_i, y - y^\mu \rangle = \sum_{i=1}^{l} p_i \left\langle d_i, \sum_{j=1}^{l} p_j d_j + \sum_{j=1}^{m} (q_j - r_j) a_j \right\rangle \leq 0. \tag{40}$$

Moreover, since $0 \leq q_i \leq C_{y^\mu} \|y - y^\mu\|$ and $0 \leq r_i \leq C_{y^\mu} \|y - y^\mu\|$, we have:

$$
\sum_{i=1}^{m} (q_i - r_i)\langle a_i, y - y^\mu \rangle \leq \left( \max_{i \in [1,m]} |\langle a_i, y - y^\mu \rangle| \right) \sum_{i=1}^{m} |q_i - r_i|
$$

$$
\leq m \left( \max_{i \in [1,m]} |\langle a_i, y - y^\mu \rangle| \right) C_{y^\mu} \|y - y^\mu\|. \tag{41}
$$

Combining Eq. (39), Eq. (40), and Eq. (41), we obtain:

$$
\|y - y^\mu\| \leq m \left( \max_{i \in [1,m]} |\langle a_i, y - y^\mu \rangle| \right) C_{y^\mu}.
$$

Moreover,

$$
\|A(y - y^\mu)\|^2 = \sum_{i=1}^{m} \langle a_i, y - y^\mu \rangle^2 \geq \left( \max_{i \in [1,m]} |\langle a_i, y - y^\mu \rangle| \right)^2.
$$

Thus, we finally get:

$$
\|A(y - y^\mu)\| \geq \frac{\|y - y^\mu\|}{m C_{y^\mu}}.
$$

Note that $C_{y^\mu}$ only depends on the set of tight constraints $I(y^\mu)$, and there are only finitely many different sets of tight constraints. Therefore, we conclude that there exists a positive constant $\beta > 0$ such that:

$$
\forall y \in \mathcal{Y} \setminus \mathcal{Y}^\mu, \ \|Ay - b^*\|^2 \geq \beta \|y - \Pi_{\mathcal{Y}^\mu}(y)\|^2
$$

$\square$

### G.7 PROOF OF LEMMA G.7

*Proof of Lemma G.7.* Let us define $h(x) := \frac{\mu}{2} \|x\|^2$. Then, we have for any $y^\mu \in \mathcal{Y}^\mu$:

$$
\max_{y' \in \mathcal{Y}} \min_{x' \in \mathcal{X}} \left( (x')^\top A y' + \frac{\mu}{2} \|x'\|^2 \right) - \min_{x' \in \mathcal{X}} \left( (x')^\top A y + \frac{\mu}{2} \|x'\|^2 \right)
$$

$$
= \min_{x' \in \mathcal{X}} \left( (x')^\top A y^\mu + \frac{\mu}{2} \|x'\|^2 \right) - \min_{x' \in \mathcal{X}} \left( (x')^\top A y + \frac{\mu}{2} \|x'\|^2 \right)
$$

$$
= - \max_{x' \in \mathcal{X}} \left( -(x')^\top A y^\mu - \frac{\mu}{2} \|x'\|^2 \right) + \max_{x' \in \mathcal{X}} \left( -(x')^\top A y - \frac{\mu}{2} \|x'\|^2 \right)
$$

$$
= h^*(-Ay) - h^*(-Ay^\mu). \tag{42}
$$

Since $h$ is a $\mu$-smooth function, the convex conjugate $h^*$ is $\frac{1}{\mu}$-strongly convex (Shalev-Shwartz, 2011, p. 149). Hence, defining $\mathcal{P} := \{-Ay \mid y \in \mathcal{Y}\}$ and $p^* := \arg\min_{p \in \mathcal{P}} h^*(p)$, we have for any $p \in \mathcal{P}$:

$$
h^*(p) - h^*(p^*) \geq \frac{1}{2\mu} \|p^* - p\|^2.
$$

Furthermore, it holds that $-Ay^\mu = \arg\min_{y \in \mathcal{Y}} h^*(-Ay^\mu) = \arg\min_{p \in \mathcal{P}} h^*(p) \Leftrightarrow -Ay^\mu = p^*$. Thus, we have:

$$
h^*(-Ay) - h^*(-Ay^\mu) \geq \frac{1}{2\mu} \|A(y^\mu - y)\|^2. \tag{43}
$$

By combining Eq. (42) and Eq. (43), we finally obtain for any $y^\mu \in \mathcal{Y}^\mu$:

$$
\max_{y' \in \mathcal{Y}} \min_{x' \in \mathcal{X}} \left( (x')^\top A y' + \frac{\mu}{2} \|x'\|^2 \right) - \min_{x' \in \mathcal{X}} \left( (x')^\top A y + \frac{\mu}{2} \|x'\|^2 \right) \geq \frac{1}{2\mu} \|A(y^\mu - y)\|^2.
$$

$\square$

## H    PROOF OF THEOREM C.1

*Proof of Theorem C.1.*  First, we prove that $y^\mu \neq y^*$ under the assumption that $\mu \neq \frac{v^* - \left(\frac{\mathbf{1}_m}{m}\right)^\top A\left(\frac{\mathbf{1}_n}{n}\right)}{\|y^*\|^2 - \frac{1}{n}}$ by contradiction. We assume that $y^\mu = y^*$. From Eq. (13), we have for any $x \in \Delta^m$:

$$x^\top A y^\mu + \frac{\mu_x}{2} \|x\|^2 = x^\top A y^* + \frac{\mu_x}{2} \|x\|^2 = v^* + \frac{\mu_x}{2} \|x\|^2. \tag{44}$$

On the other hand,

$$\left(\frac{\mathbf{1}_m}{m}\right)^\top A^\top y^\mu + \frac{\mu_x}{2} \left\|\frac{\mathbf{1}_m}{m}\right\|^2 = \left(\frac{\mathbf{1}_m}{m}\right)^\top A^\top y^* + \frac{\mu_x}{2} \left\|\frac{\mathbf{1}_m}{m}\right\|^2 = v^* + \frac{\mu_x}{2} \left\|\frac{\mathbf{1}_m}{m}\right\|^2. \tag{45}$$

By combining Eq. (44) and Eq. (45), we have for any $x \in \Delta^m$:

$$x^\top A y^\mu + \frac{\mu_x}{2} \|x\|^2 \geq \left(\frac{\mathbf{1}_m}{m}\right)^\top A^\top y^\mu + \frac{\mu_x}{2} \left\|\frac{\mathbf{1}_m}{m}\right\|^2.$$

Hence, from the property of the player $x$'s equilibrium strategy in the perturbed game, $x^\mu$ must satisfy $x^\mu = \frac{\mathbf{1}_m}{m}$.

On the other hand, from the property of the player $y$'s equilibrium strategy $y^\mu$ in the perturbed game, $y^\mu$ is an optimal solution of the following optimization problem:

$$\max_{y \in \Delta^n} \left\{ (x^\mu)^\top A y - \frac{\mu_y}{2} \|y\|^2 \right\}.$$

Let us define the following Lagrangian function $L(y, \kappa, \lambda)$ as:

$$L(y, \kappa, \lambda) = (x^\mu)^\top A y - \frac{\mu_y}{2} \|y\|^2 - \sum_{i=1}^{n} \kappa_i g_i(y) - \lambda h(y),$$

where $g_i(y) = -y_i$ and $h(y) = \sum_{i=1}^{n} y_i - 1$. Then, from the KKT conditions, we get the stationarity:

$$A^\top x^\mu - \mu_y y^\mu - \sum_{i=1}^{n} \kappa_i \nabla g_i(y^\mu) - \lambda \nabla h(y^\mu) = \mathbf{0}_n, \tag{46}$$

and the complementary slackness:

$$\forall i \in [n], \ \kappa_i g_i(y^\mu) = 0. \tag{47}$$

Since $y^\mu = y^*$ and $y^*$ is in the interior of $\Delta^n$, we have $g(y^\mu) = -y_i^\mu < 0$ for all $i \in [n]$. Thus, from Eq. (47), we have $\kappa_i = 0$ for all $i \in [n]$. Substituting this into Eq. (46), we obtain:

$$A^\top x^\mu - \mu_y y^\mu - \lambda \nabla h(y^\mu) = A^\top x^\mu - \mu_y y^\mu - \lambda \mathbf{1}_n = \mathbf{0}_n. \tag{48}$$

Hence, we have:

$$\lambda = \frac{\mathbf{1}_n^\top A^\top x^\mu - \mu_y}{n}. \tag{49}$$

Putting Eq. (49) into Eq. (48) yields:

$$y^\mu = \frac{1}{\mu_y} \left( A^\top x^\mu - \frac{\mathbf{1}_n^\top A^\top x^\mu - \mu_y}{n} \mathbf{1}_n \right) = \frac{1}{\mu_y} \left( A^\top \frac{\mathbf{1}_m}{m} - \frac{\mathbf{1}_n^\top A^\top \frac{\mathbf{1}_m}{m} - \mu_y}{n} \mathbf{1}_n \right),$$

where the second equality follows from $x^\mu = \frac{\mathbf{1}_m}{m}$. Multiplying this by $\frac{\mathbf{1}_m^\top}{m} A$, we have:

$$v^* = \frac{1}{\mu_y} \left( \frac{1}{m^2} \|A^\top \mathbf{1}_m\|^2 - \frac{1}{m^2 n} (\mathbf{1}_n^\top A^\top \mathbf{1}_m)^2 + \mu_y \frac{1}{mn} \mathbf{1}_m^\top A \mathbf{1}_n \right), \tag{50}$$

where we used the assumption that $y^\mu = y^*$ and Eq. (13). Here, we have:

$$\frac{1}{m^2}\left\|A^\top \mathbf{1}_m\right\|^2 - \frac{1}{m^2 n}(\mathbf{1}_n^\top A^\top \mathbf{1}_m)^2 + \mu_y \frac{1}{mn}\mathbf{1}_m^\top A\mathbf{1}_n$$

$$= \left\|v^*\mathbf{1}_n + A^\top \frac{\mathbf{1}_m}{m} - v^*\mathbf{1}_n\right\|^2 - \frac{1}{n}\left(\mathbf{1}_n^\top\left(v^*\mathbf{1}_n + A^\top \frac{\mathbf{1}_m}{m} - v^*\mathbf{1}_n\right)\right)^2 + \frac{\mu_y}{n}\mathbf{1}_n^\top\left(v^*\mathbf{1}_n + A^\top \frac{\mathbf{1}_m}{m} - v^*\mathbf{1}_n\right)$$

$$= n(v^*)^2 + \left\|A^\top \frac{\mathbf{1}_m}{m} - v^*\mathbf{1}_n\right\|^2 + 2v^*\mathbf{1}_n^\top\left(A^\top \frac{\mathbf{1}_m}{m} - v^*\mathbf{1}_n\right)$$

$$- n(v^*)^2 - \frac{1}{n}\left(\mathbf{1}_n^\top\left(A^\top \frac{\mathbf{1}_m}{m} - v^*\mathbf{1}_n\right)\right)^2 - 2v^*\mathbf{1}_n^\top\left(A^\top \frac{\mathbf{1}_m}{m} - v^*\mathbf{1}_n\right) + \mu_y v^* + \frac{\mu_y}{n}\mathbf{1}_n^\top\left(A^\top \frac{\mathbf{1}_m}{m} - v^*\mathbf{1}_n\right)$$

$$= \mu_y v^* + \left\|A^\top \frac{\mathbf{1}_m}{m} - v^*\mathbf{1}_n\right\|^2 - \frac{1}{n}\left(\mathbf{1}_n^\top\left(A^\top \frac{\mathbf{1}_m}{m} - v^*\mathbf{1}_n\right)\right)^2 + \frac{\mu_y}{n}\mathbf{1}_n^\top\left(A^\top \frac{\mathbf{1}_m}{m} - v^*\mathbf{1}_n\right)$$

$$= \mu_y v^* + \left(A^\top \frac{\mathbf{1}_m}{m} - v^*\mathbf{1}_n\right)^\top\left(\mathbb{I} - \frac{1}{n}\mathbf{1}_n\mathbf{1}_n^\top\right)\left(A^\top \frac{\mathbf{1}_m}{m} - v^*\mathbf{1}_n\right) + \frac{\mu_y}{n}\mathbf{1}_n^\top\left(A^\top \frac{\mathbf{1}_m}{m} - v^*\mathbf{1}_n\right).$$

$$(51)$$

Here, since $A^\top \frac{\mathbf{1}_m}{m} = \frac{\mathbf{1}_n^\top A^\top x^\mu - \mu_y}{n}\mathbf{1}_n + \mu_y y^\mu = \frac{\frac{1}{m}\mathbf{1}_m^\top A\mathbf{1}_n - \mu_y}{n}\mathbf{1}_n + \mu_y y^*$ from Eq. (48) and Eq. (49), we get:

$$\left(A^\top \frac{\mathbf{1}_m}{m} - v^*\mathbf{1}_n\right)^\top\left(\mathbb{I} - \frac{1}{n}\mathbf{1}_n\mathbf{1}_n^\top\right)\left(A^\top \frac{\mathbf{1}_m}{m} - v^*\mathbf{1}_n\right) + \frac{\mu_y}{n}\mathbf{1}_n^\top\left(A^\top \frac{\mathbf{1}_m}{m} - v^*\mathbf{1}_n\right)$$

$$= \left(\left(\frac{\frac{1}{m}\mathbf{1}_m^\top A\mathbf{1}_n - \mu_y}{n} - v^*\right)\mathbf{1}_n + \mu_y y^*\right)^\top\left(\mathbb{I} - \frac{1}{n}\mathbf{1}_n\mathbf{1}_n^\top\right)\left(\left(\frac{\frac{1}{m}\mathbf{1}_m^\top A\mathbf{1}_n - \mu_y}{n} - v^*\right)\mathbf{1}_n + \mu_y y^*\right)$$

$$+ \frac{\mu_y}{n}\mathbf{1}_n^\top\left(\left(\frac{\frac{1}{m}\mathbf{1}_m^\top A\mathbf{1}_n - \mu_y}{n} - v^*\right)\mathbf{1}_n + \mu_y y^*\right)$$

$$= \mu_y\left(\frac{\frac{1}{m}\mathbf{1}_m^\top A\mathbf{1}_n - \mu_y}{n} - v^*\right) + \frac{\mu_y^2}{n} + \mu_y\left(\frac{\frac{1}{m}\mathbf{1}_m^\top A\mathbf{1}_n - \mu_y}{n} - v^*\right)\mathbf{1}_n^\top\left(\mathbb{I} - \frac{1}{n}\mathbf{1}_n\mathbf{1}_n^\top\right)y^*$$

$$+ \mu_y\left(\frac{\frac{1}{m}\mathbf{1}_m^\top A\mathbf{1}_n - \mu_y}{n} - v^*\right)(y^*)^\top\left(\mathbb{I} - \frac{1}{n}\mathbf{1}_n\mathbf{1}_n^\top\right)\mathbf{1}_n + \mu_y^2(y^*)^\top\left(\mathbb{I} - \frac{1}{n}\mathbf{1}_n\mathbf{1}_n^\top\right)y^*$$

$$= \mu_y\left(\frac{\frac{1}{m}\mathbf{1}_m^\top A\mathbf{1}_n - \mu_y}{n} - v^*\right) + \mu_y^2\left\|y^*\right\|^2$$

$$= \mu_y\left(\mu_y\left(\left\|y^*\right\|^2 - \frac{1}{n}\right) + \left(\frac{\mathbf{1}_m}{m}\right)^\top A\left(\frac{\mathbf{1}_n}{n}\right) - v^*\right). \tag{52}$$

By combining Eq. (50), Eq. (51), and Eq. (52), we have:

$$\mu_y\left(\left\|y^*\right\|^2 - \frac{1}{n}\right) + \left(\frac{\mathbf{1}_m}{m}\right)^\top A\left(\frac{\mathbf{1}_n}{n}\right) - v^* = 0.$$

Therefore, if $y^\mu = y^*$, then $\mu_y$ must satisfy:

$$\mu_y = \frac{v^* - \left(\frac{\mathbf{1}_m}{m}\right)^\top A\left(\frac{\mathbf{1}_n}{n}\right)}{\left(\left\|y^*\right\|^2 - \frac{1}{n}\right)},$$

and this is equivalent to:

$$\mu_y \neq \frac{v^* - \left(\frac{\mathbf{1}_m}{m}\right)^\top A\left(\frac{\mathbf{1}_n}{n}\right)}{\left(\left\|y^*\right\|^2 - \frac{1}{n}\right)} \Rightarrow y^\mu \neq y^*.$$

By a similar argument, in terms of player $x$, we can conclude that:

$$\mu_x \neq \frac{\left(\frac{\mathbf{1}_m}{m}\right)^\top A\left(\frac{\mathbf{1}_n}{n}\right) - v^*}{\left(\|x^*\|^2 - \frac{1}{m}\right)} \Rightarrow x^\mu \neq x^*.$$

$\square$

# I  PROOFS FOR THEOREM D.1

## I.1  PROOF OF THEOREM D.1

*Proof of Theorem D.1.* Let us define the total number of anchoring strategy updates as $K := \frac{T}{T_\sigma}$. For $k \in [K]$ and the $k$-th anchoring strategy $\sigma^k$, we also define the stationary point $x^{\mu,k}$, which satisfies the following condition:

$$x^{\mu,k} = \arg\min_{x \in \mathcal{X}} \left\{ g(x) + \frac{\mu}{2} \left\| x - \sigma^k \right\|^2 \right\}, \tag{53}$$

From the first-order optimality condition for $x^{\mu,k}$, we can derive the following lemma for $k$ such that $x^{\mu,k}$:

**Lemma I.1.** *For any $k \in [K]$ such that $x^{\mu,k} \notin \mathcal{X}^*$, we have for any $x^* \in \mathcal{X}^*$:*

$$\frac{1}{2} \left\| x^* - \sigma^{k+1} \right\|^2 \leq \frac{1}{2} \left\| x^* - \sigma^k \right\|^2 - \frac{\alpha^2}{4\mu^2} + \frac{D^2\mu^2}{\alpha^2} \left\| x^{\mu,k} - \sigma^{k+1} \right\|^2.$$

Here, we derive the following upper bound on $\left\| x^{\mu,k} - \sigma^{k+1} \right\|^2$:

**Lemma I.2.** *For an arbitrary perturbation strength $\mu > 0$, if we use the learning rate $\eta < \min\left( \frac{\mu}{2(\mu^2 + \|A\|^2)}, \frac{8(\mu + \|A\|)}{D\min\left(\mu, \frac{\beta_k}{\mu}\right)} \right)$, then $\sigma^{k+1}$ satisfies for any $k \in [K]$:*

$$\left\| x^{\mu,k} - \sigma^{k+1} \right\|^2 \leq \frac{256 D^2 (\mu + \|A\|)^2}{\eta^2 \min\left( \mu^2, \frac{\beta_k^2}{\mu^2} \right) T_\sigma},$$

*where $\beta_k > 0$ is a positive constant depending only on $\mathcal{Y}^{\mu,k} := \arg\max_{y \in \mathcal{Y}} (x^{\mu,k})^\top A y$.*

Note that since $\beta_k \leq \|A\|^2$ for any $k \in [K]$, Lemma I.2 holds under the assumption that $\eta < \min\left( \frac{\mu}{2(\mu^2 + \|A\|^2)}, \frac{8(\mu + \|A\|)}{D\min\left(\mu, \frac{\|A\|^2}{\mu}\right)} \right)$. By combining Lemmas I.1 and I.2, we have for any $x^* \in \mathcal{X}^*$ and $k \in [K]$ such that $x^{\mu,k} \notin \mathcal{X}^*$:

$$\frac{1}{2} \left\| x^* - \sigma^{k+1} \right\|^2 \leq \frac{1}{2} \left\| x^* - \sigma^k \right\|^2 - \frac{\alpha^2}{4\mu^2} + \frac{D^2\mu^2}{\alpha^2} \frac{\gamma_k}{T_\sigma},$$

where $\gamma_k := \frac{256 D^2 (\mu + \|A\|)^2}{\eta^2 \min\left( \mu^2, \frac{\beta_k^2}{\mu^2} \right)}$. Setting $x^* = \Pi_{\mathcal{X}^*}(\sigma^k)$ in this inequality, we get:

$$\frac{1}{2} \left\| \Pi_{\mathcal{X}^*}(\sigma^{k+1}) - \sigma^{k+1} \right\|^2 \leq \frac{1}{2} \left\| \Pi_{\mathcal{X}^*}(\sigma^k) - \sigma^{k+1} \right\|^2$$

$$\leq \frac{1}{2} \left\| \Pi_{\mathcal{X}^*}(\sigma^k) - \sigma^k \right\|^2 - \frac{\alpha^2}{4\mu^2} + \frac{D^2\mu^2}{\alpha^2} \frac{\gamma_k}{T_\sigma}. \tag{54}$$

We now consider two cases depending on whether $x^{\mu,k} \notin \mathcal{X}^*$ for all $k \in [K]$.

**Case 1:** In the first case, we suppose that $x^{\mu,k} \notin \mathcal{X}^*$ for all $k \in [K]$. In this case, we can iteratively apply Eq. (54). Thus, we obtain:

$$\frac{1}{2}\left\|\Pi_{\mathcal{X}^*}(\sigma^{K+1}) - \sigma^{K+1}\right\|^2 \le \frac{1}{2}\left\|\Pi_{\mathcal{X}^*}(\sigma^1) - \sigma^1\right\|^2 - \frac{K\alpha^2}{4\mu^2} + \frac{KD^2\mu^2}{\alpha^2}\frac{\gamma}{T_\sigma},$$

where $\gamma := \max_{k \in [K]} \gamma_k$. Since $\sigma^1 = x^1$, we have:

$$\frac{1}{2}\left\|\Pi_{\mathcal{X}^*}(\sigma^{K+1}) - \sigma^{K+1}\right\|^2 \le \frac{1}{2}\left\|\Pi_{\mathcal{X}^*}(x^1) - x^1\right\|^2 - \frac{K\alpha^2}{4\mu^2} + \frac{KD^2\mu^2}{\alpha^2}\frac{\gamma}{T_\sigma},$$

Thus, if $K \ge \frac{2\mu^2\left\|\Pi_{\mathcal{X}^*}(x^1)-x^1\right\|^2}{\alpha^2}$,

$$\frac{1}{2}\left\|\Pi_{\mathcal{X}^*}(\sigma^{K+1}) - \sigma^{K+1}\right\|^2 \le \frac{D^2\mu^2\gamma K}{\alpha^2 T_\sigma}.$$

**Case 2:** Next, we consider the case where there exists $k \in [K]$ such that $x^{\mu,k} \in \mathcal{X}^*$. Defining $k_{\max} := \max_{k \in [K]}\left\{k \mid x^{\mu,k} \in \mathcal{X}^*\right\}$, we have $x^{\mu,k} \notin \mathcal{X}^*$ for any $k \in \{k_{\max}+1, \cdots, K\}$. Hence, we can apply Eq. (54) from $k = K, K-1, \cdots, k_{\max}+1$.

$$\frac{1}{2}\left\|\Pi_{\mathcal{X}^*}(\sigma^{K+1}) - \sigma^{K+1}\right\|^2 \le \frac{1}{2}\left\|\Pi_{\mathcal{X}^*}(\sigma^{k_{\max}+1}) - \sigma^{k_{\max}+1}\right\|^2 - \frac{(K-k_{\max})\alpha^2}{4\mu^2} + \frac{(K-k_{\max})D^2\mu^2}{\alpha^2}\frac{\gamma}{T_\sigma}.$$

Since $\Pi_{\mathcal{X}^*}(x^{\mu,k_{\max}}) = x^{\mu,k_{\max}}$ in this case, it holds that:

$$\frac{1}{2}\left\|\Pi_{\mathcal{X}^*}(\sigma^{k_{\max}+1}) - \sigma^{k_{\max}+1}\right\|^2 \le \frac{1}{2}\left\|\Pi_{\mathcal{X}^*}(x^{\mu,k_{\max}}) - \sigma^{k_{\max}+1}\right\|^2 = \frac{1}{2}\left\|x^{\mu,k_{\max}} - \sigma^{k_{\max}+1}\right\|^2,$$

Combining these two inequalities, we have:

$$\frac{1}{2}\left\|\Pi_{\mathcal{X}^*}(\sigma^{K+1}) - \sigma^{K+1}\right\|^2 \le \frac{1}{2}\left\|x^{\mu,k_{\max}} - \sigma^{k_{\max}+1}\right\|^2 - \frac{(K-k_{\max})\alpha^2}{4\mu^2} + \frac{(K-k_{\max})D^2\mu^2}{\alpha^2}\frac{\gamma}{T_\sigma}.$$

By applying Lemma I.2 to this inequality, we have:

$$\begin{aligned}\frac{1}{2}\left\|\Pi_{\mathcal{X}^*}(\sigma^{K+1}) - \sigma^{K+1}\right\|^2 &\le \frac{\gamma}{T_\sigma} - \frac{(K-k_{\max})\alpha^2}{4\mu^2} + \frac{(K-k_{\max})D^2\mu^2}{\alpha^2}\frac{\gamma}{T_\sigma}\\ &\le \frac{\gamma}{T_\sigma} + \frac{D^2\mu^2\gamma K}{\alpha^2 T_\sigma} = \gamma\left(1 + \frac{D^2\mu^2}{\alpha^2}\right)\frac{K}{T_\sigma}.\end{aligned}$$

Therefore, in either case, if $K \ge \frac{2\mu^2\left\|\Pi_{\mathcal{X}^*}(x^1)-\sigma^1\right\|^2}{\alpha^2}$, then we have:

$$\frac{1}{2}\left\|\Pi_{\mathcal{X}^*}(\sigma^{K+1}) - \sigma^{K+1}\right\|^2 \le \gamma\left(1 + \frac{D^2\mu^2}{\alpha^2}\right)\frac{K}{T_\sigma} = \gamma\left(1 + \frac{D^2\mu^2}{\alpha^2}\right)\frac{T}{T_\sigma^2}. \tag{55}$$

Setting $T_\sigma = cT$ for a constant $c \le \min\left(1, \frac{\alpha^2}{2\mu^2\|\Pi_{\mathcal{X}^*}(\sigma^1)-\sigma^1\|^2}\right)$, it holds that $K = \frac{T}{T_\sigma} = \frac{1}{c} \ge \frac{2\mu^2\left\|\Pi_{\mathcal{X}^*}(\sigma^1)-\sigma^1\right\|^2}{\alpha^2}$. Therefore, the condition for Eq. (55) is satisfied, and we obtain:

$$\frac{1}{2}\left\|\Pi_{\mathcal{X}^*}(\sigma^{K+1}) - \sigma^{K+1}\right\|^2 \le \gamma\left(1 + \frac{D^2\mu^2}{\alpha^2}\right)\frac{1}{cT}.$$

Since $\sigma^{K+1} = x^{T_\sigma K+1} = x^{T+1}$, we have finally:

$$\frac{1}{2}\left\|\Pi_{\mathcal{X}^*}(x^{T+1}) - x^{T+1}\right\|^2 \le \gamma\left(1 + \frac{D^2\mu^2}{\alpha^2}\right)\frac{1}{cT}.$$

$\square$

## I.2 PROOF OF LEMMA I.1

*Proof of Lemma I.1.* Since $g$ is a convex function on $\mathcal{X}$ (Boyd & Vandenberghe, 2004), we can utilize the first-order optimality condition for $x^{\mu,k}$. Specifically, there exists a subgradient $h^{k+1} \in \partial g(x^{\mu,k})$ such that:

$$\forall x \in \mathcal{X}, \; \langle h^{k+1} + \mu(x^{\mu,k} - \sigma^k), x - x^{\mu,k} \rangle \geq 0.$$

From the definition of the subgradient, we have for any $x \in \mathcal{X}$:

$$\langle h^{k+1}, x^{\mu,k} - x \rangle \geq g(x^{\mu,k}) - g(x^*).$$

Combining these inequalities, we get for any $x \in \mathcal{X}$:

$$g(x^{\mu,k}) - g(x) \leq \mu \langle x^{\mu,k} - \sigma^k, x - x^{\mu,k} \rangle.$$

By taking $x = x^*$ for an arbitrary $x^* \in \mathcal{X}^*$, we obtain:

$$g(x^{\mu,k}) - v^* = g(x^{\mu,k}) - g(x^*) \leq \mu \langle x^{\mu,k} - \sigma^k, x - x^{\mu,k} \rangle. \tag{56}$$

On the other hand, from Eq. (23), we have for any $k \geq 1$:

$$\frac{1}{2} \left\| x^* - x^{\mu,k} \right\|^2 - \frac{1}{2} \left\| x^* - \sigma^k \right\|^2 + \frac{1}{2} \left\| x^{\mu,k} - \sigma^k \right\|^2 = \left\langle x^{\mu,k} - \sigma^k, x^{\mu,k} - x^* \right\rangle. \tag{57}$$

By combining Eq. (56) and Eq. (57), we get:

$$\frac{1}{2} \left\| x^* - x^{\mu,k} \right\|^2 - \frac{1}{2} \left\| x^* - \sigma^k \right\|^2 + \frac{1}{2} \left\| x^{\mu,k} - \sigma^k \right\|^2 \leq \frac{v^* - g(x^{\mu,k})}{\mu} \leq 0.$$

Thus, from Cauchy-Schwarz inequality and Young's inequality, we have for any $\rho > 0$:

$$\frac{1}{2} \left\| x^{\mu,k} - \sigma^k \right\|^2$$

$$\leq \frac{1}{2} \left\| x^* - \sigma^k \right\|^2 - \frac{1}{2} \left\| x^* - x^{\mu,k} \right\|^2$$

$$\leq \frac{1}{2} \left\| x^* - \sigma^k \right\|^2 - \frac{1}{2} \left\| x^* - \sigma^{k+1} + \sigma^{k+1} - x^{\mu,k} \right\|^2$$

$$\leq \frac{1}{2} \left\| x^* - \sigma^k \right\|^2 - \frac{1}{2} \left\| x^* - \sigma^{k+1} \right\|^2 - \frac{1}{2} \left\| \sigma^{k+1} - x^{\mu,k} \right\|^2 - \left\langle x^* - \sigma^{k+1}, \sigma^{k+1} - x^{\mu,k} \right\rangle$$

$$\leq \frac{1}{2} \left\| x^* - \sigma^k \right\|^2 - \frac{1}{2} \left\| x^* - \sigma^{k+1} \right\|^2 - \frac{1}{2} \left\| \sigma^{k+1} - x^{\mu,k} \right\|^2 + \frac{\rho}{2} \left\| \sigma^{k+1} - x^{\mu,k} \right\|^2 + \frac{1}{2\rho} \left\| x^* - \sigma^{k+1} \right\|^2.$$

Hence,

$$\frac{1}{2} \left\| x^* - \sigma^{k+1} \right\|^2 \leq \frac{1}{2} \left\| x^* - \sigma^k \right\|^2 - \frac{1}{2} \left\| x^{\mu,k} - \sigma^k \right\|^2 - \frac{1}{2} \left\| \sigma^{k+1} - x^{\mu,k} \right\|^2$$

$$+ \frac{\rho}{2} \left\| \sigma^{k+1} - x^{\mu,k} \right\|^2 + \frac{1}{2\rho} \left\| x^* - \sigma^{k+1} \right\|^2. \tag{58}$$

Here, we derive the lower bound on the distance between $\sigma^k$ and $x^{\mu,k}$:

**Lemma I.3.** *For any $k \in [K]$ such that $x^{\mu,k} \notin \mathcal{X}^*$, we have:*

$$\frac{\alpha}{\mu} \leq \left\| x^{\mu,k} - \sigma^k \right\|.$$

By combining Eq. (58) and Lemma I.3, we have for any $k \in [K]$ such that $x^{\mu,k} \notin \mathcal{X}^*$:

$$\frac{1}{2} \left\| x^* - \sigma^{k+1} \right\|^2 \leq \frac{1}{2} \left\| x^* - \sigma^k \right\|^2 - \frac{\alpha^2}{2\mu^2} + \frac{\rho}{2} \left\| \sigma^{k+1} - x^{\mu,k} \right\|^2 + \frac{1}{2\rho} \left\| x^* - \sigma^{k+1} \right\|^2$$

$$\leq \frac{1}{2} \left\| x^* - \sigma^k \right\|^2 - \frac{\alpha^2}{2\mu^2} + \frac{\rho}{2} \left\| \sigma^{k+1} - x^{\mu,k} \right\|^2 + \frac{D^2}{2\rho}.$$

Taking $\rho = \frac{2D^2\mu^2}{\alpha^2}$, we finally have for any $x^* \in \mathcal{X}^*$ and $k \in [K]$ such that $x^{\mu,k} \notin \mathcal{X}^*$:

$$\frac{1}{2} \left\| x^* - \sigma^{k+1} \right\|^2 \leq \frac{1}{2} \left\| x^* - \sigma^k \right\|^2 - \frac{\alpha^2}{4\mu^2} + \frac{D^2\mu^2}{\alpha^2} \left\| \sigma^{k+1} - x^{\mu,k} \right\|^2.$$

$$\square$$

### I.3   PROOF OF LEMMA I.2

*Proof of Lemma I.2.* By applying the same proof technique as in Theorem 4.1, under the assumption that $\eta < \min\left(\frac{\mu}{2(\mu^2+\|A\|^2)}, \frac{8(\mu+\|A\|)}{D\min\left(\mu, \frac{\beta_k}{\mu}\right)}\right)$, we have for any $k \in [K]$ and $t \in \{(k-1)T_\sigma + 1, \cdots, kT_\sigma\}$:

$$\left\|x^{\mu,k} - x^{t+1}\right\|^2 \leq \frac{256D^2(\mu+\|A\|)^2}{\eta^2 \min\left(\mu^2, \frac{\beta_k^2}{\mu^2}\right)(t-(k-1)T_\sigma+1)},$$

where $\beta_k > 0$ depends only on the set $\mathcal{Y}^{\mu,k} := \arg\max_{y\in\mathcal{X}}(x^{\mu,k})^\top Ay$.

Taking $t = kT_\sigma$, we have $x^{t+1} = x^{kT_\sigma+1} = \sigma^{k+1}$ from the definition of $\sigma^{k+1}$. Thus, we have:

$$\left\|x^{\mu,k} - \sigma^{k+1}\right\|^2 \leq \frac{256D^2(\mu+\|A\|)^2}{\eta^2 \min\left(\mu^2, \frac{\beta_k^2}{\mu^2}\right)(T_\sigma+1)} \leq \frac{256D^2(\mu+\|A\|)^2}{\eta^2 \min\left(\mu^2, \frac{\beta_k^2}{\mu^2}\right)T_\sigma}.$$

$\square$

### I.4   PROOF OF LEMMA I.3

*Proof of Lemma I.3.* By taking $x^* = \Pi_{\mathcal{X}^*}(x^{\mu,k})$ in Eq. (56), we have:

$$g(x^{\mu,k}) - v^* \leq \mu\left\langle x^{\mu,k} - \sigma^k, \Pi_{\mathcal{X}^*}(x^{\mu,k}) - x^{\mu,k}\right\rangle.$$

By combining this inequality and Lemma F.1, we get:

$$\alpha\left\|x^{\mu,k} - \Pi_{\mathcal{X}^*}(x^{\mu,k})\right\| \leq \mu\left\langle x^{\mu,k} - \sigma^k, \Pi_{\mathcal{X}^*}(x^{\mu,k}) - x^{\mu,k}\right\rangle$$
$$\leq \mu\left\|x^{\mu,k} - \sigma^k\right\|\left\|\Pi_{\mathcal{X}^*}(x^{\mu,k}) - x^{\mu,k}\right\|.$$

Therefore, as long as $x^{\mu,k} \notin \mathcal{X}^*$, it holds that:

$$\frac{\alpha}{\mu} \leq \left\|x^{\mu,k} - \sigma^k\right\|.$$

$\square$

