# OpenReview forum: "Asymmetric Perturbation in Solving Bilinear Saddle-Point Optimization"
_ICLR.cc/2026/Conference — ICLR 2026 Conference Withdrawn Submission_

### Official Review · Reviewer_h63h · 2025-10-25

**Soundness:** 3
**Presentation:** 3
**Contribution:** 2
**Rating:** 2
**Confidence:** 5

**Summary:**

This paper studies matrix games that are solved via online learning. The authors study a one-sided perturbation / regularization of the form $\min_{x\in \Delta} \max_{y\in \Delta} x^\top Ay + \mu\|x\|_2^2$. Most prior work in online learning studies perturbation applied to both the $x$ and $y$ players. The authors show that this perturbation does not change the Nash equilibrium at all, for sufficiently small $\mu$. Then, they study the application of gradient descent-ascent applied to the perturbed problem. The authors show last-iterate convergence of this algorithm, at a rate of $O(\|A\|^2/T\mu^2)$. Then they apply the same idea to extensive-form games, though without any theoretical results, and run it numerically against some of the best competing algorithms from 2019 and earlier.

**Strengths:**

- The observation that the Nash equilibrium is not changed for small amounts of asymmetric perturbation is nice; I have not seen this observation before, and it is definitely neat.
- The overall topic of the paper is interesting

**Weaknesses:**

- The approach is very similar in flavor to Nesterov smoothing (which constructs the same approximation), yet Nesterov smoothing is not discussed at all. The difference between the two approaches are that Nesterov smoothing applies an accelerated method to the non-smoothed player's objective, which is now smooth, whereas the present paper applies GDA to each of the two players. The rate achieved by Nesterov smoothing is $O(1/T)$, by choosing the smoothing parameter $\mu$ optimally relative to the number of steps. The present paper achieves a $O(1/(T\mu^2))$ rate, which is significantly worse, since $\mu$ needs to be relatively small in order for the guarantees to be interesting. Also, the paper claims that "asymmetric perturbation" is novel, but it's really not; it was studied *before* symmetric perturbation, from the perspective of Nesterov smoothing.
- Once we fix $\mu$ and apply asymmetric perturbation, the natural question is: what algorithm should one apply? The authors apply GDA, and achieve $O(\|A\|/T\mu^2)$ convergence. However, it is already known from [a]  that, if we accept ergodic rates rather than last iterate, then we can get $O(\|A\|/T^2\mu)$, which is substantially better, both in terms of dependence on $\mu$ and $T$. Of course, the authors are focusing on last iterate, so it is possible that we must accept a slower rate, but the dependence on both $\|A\|$ and $\mu$ seems pretty bad in this work. Especially considering that you usually get a better dependence on $\|A\|$ with non-accelerated FOMs / non-optimistic learning dynamics.
- The experiments leave out the PCFR+ algorithm, as well as follow-up work on PCFR+, from their experiments. This is unfortunate, because PCFR+ is much faster than the best competing algorithm in many of the benchmark games chosen by the authors. Thus, the authors only compare to algorithms from 2019 and earlier, which leaves out about 6 years of progress! In my opinion this is an egregious oversight (especially because it's an oversight that conveniently lets the authors claim that their algorithm is the best), and the paper cannot be accepted in its current form.

**Questions:**

No questions.

---

### Official Review · Reviewer_SFtf · 2025-10-27

**Soundness:** 3
**Presentation:** 3
**Contribution:** 2
**Rating:** 6
**Confidence:** 3

**Summary:**

This paper studies properties of minmax problems with perturbed (regularized) objectives, and algorithms for obtaining last-iterate convergence to their Nash equilibria. The focus is on the bilinear case with asymmetric perturbations, where only one of the player's objective function is perturbed with a euclidean regularizer. The main results of the paper establish the following:

1.  When the perturbation weight is a sufficiently small constant, the minimax strategy of the perturbed game coincides with the minimax strategy of the original game.

2. Running alternating GDA on the perturbed objective leads to last-iterate convergence to a minimax strategy of the original game at a rate of 1/T.

The authors also give some empirical results demonstrating the effectiveness of asymmetric perturbations in extensive-form games.

**Strengths:**

+ In general the paper is well presented and the main technical ideas are easy to follow.
+ Obtaining a last-iterate convergence rate in the bilinear zero-sum game setting *without* optimism is significant, despite the fact that the O(1/T) rate in this setting is suboptimal (compared to OGDA).

**Weaknesses:**

+ The "equilibrium invariance" property of Theorem 3.1 requires a sufficiently small perturbation weight \mu, and the upper bound on \mu depends on some game-dependent constant, which in general could be very small. (See also several of the questions below).

+ The main equilibrium invariance and last-iterate convergence results are limited to the asymmetric perturbation setting using the squared euclidean norm regularizer. This raises the question of whether similar results could be established for other strongly-convex regularizers such as negative entropy.

**Questions:**

Q1 -- regarding Theorem 3.1 and the constants \mu and \alpha:
* Could you provide some concrete example regarding the constant \alpha? For example, on the biased RPS game from Figure 3, what are some example values of \alpha (and thus the corresponding allowable range for \mu?)

Q2 -- beyond euclidean regularization:
* Can you mention whether similar invariance and last-iterate convergence results are likely to hold beyond the (asymmetric) perturbations with squared l2 norm? For example, if the perturbation is negative entropy?

Q3 -- convergence of AsympP-GDA with simultaneous updates:
* As mentioned in footnote 3 and equation (6), the AsymP-GDA algorithm uses alternating udpates. Does the last-iterate convergence result still hold under *simultaneous* updates?

Q4 -- regarding Corollary 4.2:
* The corollary as stated gives a last-iterate rate for the x-player. Can you simultaneously guarantee any convergence rate for the y-player? In other words, while you mentioned that to find a maximin strategy you can solve the perturbed game with respect to y; But supposing you simply run the update rule in expression (6). Can you say anything about the convergence of \{y_t\} ?

---

### Official Review · Reviewer_SKW9 · 2025-11-01

**Soundness:** 3
**Presentation:** 3
**Contribution:** 2
**Rating:** 2
**Confidence:** 4

**Summary:**

The paper introduces a new algorithm for computing an equilibrium of a zero-sum game. Unlike standard earlier approaches, it is based on an asymmetric perturbation whereby a strongly convex regularizer is introduced only to one of the player's utility. It shows that the algorithm converges to an exact equilibrium in a last-iterate sense. Furthermore, it combines the new algorithm with the CFR framework to develop a new algorithm for solving zero-sum games in extensive form. The experiments show promise relative to other common algorithms used in the literature.

**Strengths:**

The paper examines an important problem at the heart of game theory and optimization---namely, solving zero-sum games. The proposed method is, to my knowledge, new and fairly natural. A standard approach is to add a perturbation to both players; this modification proposed by the paper has been unexplored. It is also interesting to see that the proposed method performs very well in extensive-form games, in fact much better than the symmetric counterpart. This is a surprising finding. The paper is also well written and it places its contributions accurately in the context of existing work.

**Weaknesses:**

On the negative side, the paper has some significant limitations and issues. First of all, I don't really agree with the main theoretical claim. The whole premise of the theory is that a symmetric perturbation only leads to an approximate equilibrium while an asymmetric one leads to an exact equilibrium. I don't see that. A symmetric perturbation leads arbitrarily close to an equilibrium. Furthermore, using standard LP arguments, there is an equilibrium whose bit complexity is polynomial in the description of the game. This means that an approximate equilibrium after a certain precision will be exact. So I believe that the symmetric perturbation also gives the same result. From a theoretical standpoint, I don't see any particular merit to the proposed method. Of course, Figure 3 is unconvincing: the reason why the dynamics in the middle figure haven't converged to an exact equilibrium is because the perturbation parameter is not small enough. The paper also makes the claim that the symmetric perturbation requires carefully tuning a certain hyperparameter; but isn't that also the case with the proposed method? This theorem only kicks in when the perturbation is small enough; again, I don't see the merit of the proposed approach compared to using symmetric perturbation. On top of that, the main result requires the perturbation to be potentially extremely small even in tiny games.

I also have concerns regarding the experimental evaluation. The paper fails to compare against the state of the art algorithm in extensive-form games, which is PCFR+ with alternation. I would expect to see experiments with PCFR+ with alternation, both using something like quadratic averaging and the last-iterate. If the proposed method is competitive with that algorithm, it would be a strong empirical finding.

**Questions:**

I have two main questions:

- Can the authors explain the theoretical merit of the proposed approach compared to symmetric perturbation? As I said above, an exact equilibrium, I believe, can be obtained via the symmetric perturbation.
- Can the authors include in the experiments PCFR+ (predictive CFR+ due to Farina et al. 2021) with both quadratic averaging and the last iterate? How does it compare against the proposed method?

---

### Official Review · Reviewer_BojH · 2025-11-01

**Soundness:** 3
**Presentation:** 3
**Contribution:** 2
**Rating:** 6
**Confidence:** 3

**Summary:**

The paper considers the bilinear saddle point problem, which is a classical and important theoretical setting in game theory and optimization. Many classes of methods have been proposed to solve this problem to avoid the challenge of non-convergence to Nash equilibria in the last-iterate sense. This paper lies in the class of perturbing the agents' objectives. However, contrary to traditional perturbing mechanisms which perturb both the minimizing and maximizing agents' objectives, this paper proposes the idea of asymmetrically perturbing only one agent's objectives. The papers theoretically shows that  any equilibrium of the corresponding game is still an equilibrium of the original game for small perturbations. Further, the paper shows that (alternating) GDA on the asymmetric game has a $\mathcal{O}(\frac{1}{t})$ last iterate convergence rate. The authors complement this theory for the matrix game setting by analyzing a asymmetrically perturbed CFR variant for extensive form games, which empirically demonstrates favorable performance.

**Strengths:**

1. The asymmetric perturbation modification introduced is quite simple and elegant and yields the desired convergence behavior.
2. Many zero-sum methods do give similar guarantees, but implementing them in practice requires a painful tuning of some parameters. In contrast,  I really appreciate the ability of not having to tune a parameter to use AsymPGDA which the paper proposes.

**Weaknesses:**

1. The proposed approach can be viewed as an instance of a more general framework where both the agents' objective can be perturbed, and these perturbations need not necessarily be equal for both agents.  My only concern is that given the classical nature of the setting, and the vast literature on perturbations in minimal optimization/game theory in general, whether such a framework already exists? To be fair to the authors, this field has received so much attention that I acknowledge that answering this question with certainty is not really possible.

**Questions:**

I think the paper was fairly clear to follow and I have no questions. I have some suggestions.

One perspective on zero sum games is through the lens of dynamical systems. I encourage the authors to investigate whether their listed future directions can benefit from such a perspective. A seminal paper in this field which investigates issues faced by gradient based algorithms to  converge to Nash equilibria in zero-sum games (particularly, GDA) is  Mazumdar, Eric, Lillian J. Ratliff, and S. Shankar Sastry. "On gradient-based learning in continuous games." SIAM Journal on Mathematics of Data Science 2.1 (2020): 103-131. As the authors are essentially deploying a two-timescale version of  GDA, I believe they might benefit from analyzing the underlying dynamical system resulting from two timescale GDA applied to the perturbed objectives.

(*To be clear:* I do not expect any answers from their end on this during the rebuttal - this is just a suggestion for investigation shall they wish to look into it in the future)

---

### Note · Authors · 2025-11-21

I have read and agree with the venue's withdrawal policy on behalf of myself and my co-authors.